



# Declining suspended sediment in United States rivers and streams: Linking sediment trends to changes in land use/cover, hydrology and climate

Jennifer C. Murphy[1]

[1] U.S. Geological Survey, Lower Mississippi-Gulf Water Science Center, Nashville, Tennessee, USA

*Correspondence to*: Jennifer C. Murphy (jmurphy@usgs.gov)

**Abstract.** Between 1992 and 2012, concentrations of annual mean suspended sediment decreased at over half (58%) of the 137 stream sites assessed across the contiguous United States (US). Increases occurred at 17% of the sites and the direction of change was uncertain at the remaining 25%. Sediment trends were characterized using the Weighted Regressions on

Time, Discharge, and Season model, and decreases in sediment ranged from -95% to -8.5% of the 1992 concentration. To explore potential drivers of these changes, the sediment trends were (1) parsed into two broad contributors of change, changes in land management versus changes in the streamflow regime, and (2) grouped by land use of the watershed and correlated to concurrent changes in land use/cover, hydrology and climate and static/long-term watershed characteristics. At 83% of the sites, changes in land management (captured by changes in the concentration-streamflow relationship over time)

contributed more to the change in the sediment trend than changes in the streamflow regime alone (i.e. any systematic change in the magnitude, frequency or timing of flows). However, at >60% of the sites, changes in the streamflow regime contributed at least a 5% change in sediment and at 10 sites changes in the streamflow regime contributed over half the change in sediment, indicating that at many sites changes in streamflow were not the main driver of changes in sediment but was often an important supporting factor. Correlations between sediment trends and concurrent changes in land use/cover,

hydrology and climate were often stronger at sites draining watersheds with more homogenous, human-related land uses (i.e. agricultural and urban lands) compared to mixed-use or undeveloped lands. At many sites, decreases in sediment occurred despite small to moderate increases in the amount of urban or agricultural land in the watershed, suggesting conservation efforts to reduce sediment runoff to streams may be successful, up to a point, even as lands are converted to urban and agricultural uses.

## 1 Introduction

Across the United States (US) and the world, sediment is one of the leading pollutants in rivers and streams (USEPA, 2008-2016; Walling, 2009), degrading aquatic habitats and affecting water usability (Brown and Froemke, 2012; Wohl, 2015).



River monitoring programs of sediment are typically implemented to collect data to characterize status and temporal changes in the delivery of suspended material, often with an explicit goal of capturing improvements. An implicit goal of many of these programs is better understanding of why sediment delivery has or has not varied over time (Irvine et al., 2015). To optimize the ability to characterize and detect temporal changes, many monitoring programs focus on implementing the best

sampling design. A missing piece is often the observation and characterization of potential causes of these changes in sediment (Irvine et al., 2015), such as shifts in land use or land cover (land use/cover), changes in management of the landscape or stream, or climatic variability.

There are multiple approaches for linking changes in sediment at stream sites with changes in land use/cover, hydrology and

climate. These approaches include using qualitative statements with or without data (e.g. Gao et al., 2013; Kreiling and Houser, 2016; Li et al., 2016), using process-based watershed and landscape models (e.g. Ficklin et a., 2013; Lacher et al., 2019) and teasing apart water-quality and streamflow records to identify and estimate the amount of change due to human actions or climate and hydrology (e.g. Wu et al., 2012; Gao et al., 2013; Li et al., 2016; Murphy and Sprague, 2019; Choquette et al., 2019; Rossi et al., 2009). Many of these approaches are hindered by the lack of available data that

characterize potential drivers of change. Such data are often not available or not available across many sites. Some studies have used geospatial data to generate estimates of various land-use and land-disturbance metrics and have been successful at linking these to spatial variations in static water-quality conditions (e.g. Mehaffey et al., 2005; Carey et al., 2011). This geospatial approach compares static land-use conditions (either current or long-term average conditions) to recent water-quality conditions and provides information about the spatial variability of water quality across many sites; however, this

approach does not explicitly explore how temporal changes in land use/cover or other human activities affect water quality. Other studies have begun to explore the effect of temporal changes on water quality by explicitly considering land-use/cover, land-management and hydrologic changes over time using hybrid deterministic-empirical approaches (Chanat and Yang, 2018), structural equation models (Ryberg, 2017; Ryberg et al., 2018) or focusing on a couple of specific potential causes in a limited geographic area (Schottler et al., 2014; Panthi et al., 2017). Historically, field-based assessments in specific areas

have been successful in identifying and supporting causal understanding of changes in sediment (e.g. Wolman and Schick, 1967; Trimble and Lund, 1982; Gellis et al., 1991).

Due to the sensitivity of sediment to streamflow conditions, concentrations (or loads) of annual mean sediment covary with annual streamflow conditions at many sites. Much of this year-to-year variability in streamflow is dependent on weather,

though at some locations there may also be a longer-term systematic change in streamflow that also influences sediment. Thus, for a given year, annual mean concentrations of sediment are a function of the streamflow conditions for that year, changes that occurred in the basin (i.e. land-management activities, surface or channel disturbance, etc.), and, at some locations, a systematic change in some portion of the streamflow regime (Murphy and Sprague, 2019; Choquette et al.,





2019). When trying to understand potential drivers of long-term changes in sediment, it is the latter two contributions that are of most interest.

All rivers have a characteristic streamflow regime that captures the typical pattern of fluctuations in the magnitude, timing
and frequency of streamflow across a given year. Under stationary climate, a streamflow regime is also often stationary; however, natural and anthropogenic influences, such as increases in precipitation or a change in dam operations, can cause changes in the streamflow regime over time. These changes occur in many forms, such as increases in mean streamflow, decreases in high streamflow events or a shift in high streamflow from spring to winter. Changes in the streamflow regime may ultimately lead to changes in sediment concentrations in a stream because of shifts in transport processes or changes in
which channel or near-channel sediment sources are eroded. This can include geomorphological changes like increased headcutting and channel bank sloughing, as well as channel bottom scouring and resuspension. The effect of streamflow-related changes on sediment can be compared to changes in sediment resulting from management and disturbance on the landscape across the watershed. Changes in landscape management (including surface disturbance and other human actions in the watershed) may enhance or aim to minimize the sediment available for transport to a stream via overland runoff. For
example, changes in the amount of land used for grazing or crops, changes in tillage practices, increased construction in suburban or exurban areas, increases in mining or harvesting of timber can all lead to enhanced erosion. However, other management actions, such as widespread implementation of agricultural or urban best-management practices (BMPs), enrolling agricultural land in the Conservation Reserve Program (CRP) and channel restoration efforts, are aimed at reducing erosion or trapping sediment. Attributing sediment trends to these broad categories of change (i.e. landscape management
versus the streamflow regime) provides promise for better understanding the relative influence of largely controllable human influences on sediment in streams, resulting from changes in land management and surface disturbance, compared to the influence of less controllable changes in the streamflow regime.

In this paper, an extensive dataset of temporal changes in land use/cover, hydrology and climate (Falcone, 2017; Falcone et
al., 2019; Farmer et al., 2017) is used in conjunction with two decades of sediment and streamflow data (De Cicco et al., 2017; Oelsner et al., 2017) to characterize changes in annual mean concentrations of suspended sediment (hereafter referred to as sediment) and explore potential drivers of these changes at 137 stream sites across the contiguous US. The objectives are to (1) summarize and describe sediment trends between 1992 and 2012, (2) explore contributions to sediment trends from changes in landscape management versus changes in the streamflow regime and (3) link specific land-use/cover and hydro-
climatic changes (across the watershed and for a near-site, near-stream "proximal" zone) and static and long-term watershed characteristics to sediment trends. This paper builds off the insights presented in Murphy and Sprague (2019) by explicitly exploring contributions from changes in landscape management versus changes in the streamflow regime for sediment, regionally and by land use. It also goes beyond the analysis presented in Murphy and Sprague (2019) by exploring changes



in overall sediment across the US, how these changes vary regionally and with land use, and links these changes in sediment to observed changes in land use/cover, streamflow and climate. The overarching goals of this effort are to better understand how sediment concentration has changed over time across the US and to provide insight into the potential drivers of these trends.

## 2.0 Methods

### 2.1 Description of water-quality data and trend results

This study relied on the water-quality data and trend analyses described in Oelsner et al. (2017) - a comprehensive trend assessment of US rivers for multiple water-quality and ecologic metrics, spanning four trend periods beginning as early as

1972. With nearly 12,000 reported trend results for approximately 1,500 sites, the focus of their publication was to document data acquisition, harmonization and screening processes and the trend analysis methods. The trend results and data were published in De Cicco et al. (2017) and Murphy et al. (2018). The data originated from 74 Federal, state and local governments and organizations that collect and process stream water-quality samples across the contiguous US. Each site was associated with a streamflow gage. For this paper, the 1992 to 2012 sediment trends (annual estimates and changes)

were extracted from Murphy et al. (2018) and sites with drainage areas < 300,000 square kilometers (sq km) were retained; 7 sites that had very large drainages ranging from 410,000 sq km to 1,080,000 sq km were excluded. An additional site, on the Atchafalaya River in Louisiana (site number 07381495), was also excluded because a large proportion of the water at this site is diverted from an adjacent drainage basin. The extracted sediment trends included results for suspended-sediment concentration (SSC) and total suspended solids concentration (TSS), at 99 sites and 41 sites, respectively. A few sites (n = 3)

had data and trends for both parameters, resulting in 137 unique sites overall.

SSC and TSS characterize suspended material in the river column, but these estimates are not directly comparable and must be interpreted somewhat differently. ASTM Standard Test Method D 3977-97 was used for SSC determinations (American Society for Testing and Materials, 2000) and Method 2540 D, with some variations (Gray et al., 2000), was used for TSS

determinations (American Public Health Association, American Water Works Association, and Water Pollution Control Federation, 1995). The difference in SSC and TSS determinations is largely due to differences in the water-sample preparation procedures, resulting in different suspended particle-size distributions for the same water sample. The comparability of these sediment parameters is described in Gray et al. (2000) and briefly summarized here. SSC is determined by measuring the dry weight of all sediment from a water sample of a known volume. Several techniques are

used to determine TSS and most techniques similarly measure the dry weight of all sediment from a water sample of a known volume; however, this technique as defined by the TSS protocol (American Public Health Association, American Water Works Association, and Water Pollution Control Federation, 1995) weighs the sediment in only a 100-milliter sub-



sample from the original water sample. Due to the physical properties of sediment and water, taking an aliquot of the original water sample tends to leave larger particle sizes (often sands) in the original sample. Thus, TSS generally characterizes only finer suspended particle sizes, whereas SSC characterizes the entire suspended particle-size distribution of the original sample and presumably the river. The downward bias of TSS compared to SSC, especially at sites with larger proportions of

sand-size sediment, is an important consideration when interpreting changes in TSS or comparing TSS to SSC. Both parameters are reported here because SSC determinations are more accurate, reliable and presumably characterize the entire suspended particle-size distribution of the sampled stream, whereas TSS determinations are much more common across the US.

Oelsner et al. (2017) and Murphy et al. (2018) provide a complete description of the modelling specifications used to generate trends presented in this study. Briefly, the Weighted Regressions on Time, Discharge and Season model (WRTDS; Hirsch et al., 2010; Hirsch et al., 2018a; Choquette et al., 2019) was used to calculate trends between 1992 and 2012. For each site, WRTDS estimates daily mean concentrations using weighted regression.  These estimated daily concentrations are flow normalized (FN) to remove the influence of year-to-year variability from streamflow, which is mostly weather-driven,

and the non-FN and FN daily estimates are separately aggregated to non-FN and FN annual mean concentrations. Flow normalization is a physically based smoothing technique that uses the observed streamflow values to provide an estimate of concentration that excludes effects from random year-to-year fluctuations in streamflow but retains the effects from both seasonal streamflow variability and long-term, systematic streamflow trends (Choquette et al., 2019). Trends are reported as the time series of FN annual values and as the change (in both milligrams per liter (mg/L) and percent change relative to

initial concentrations) between the 1992 and 2012 FN sediment concentration. Thus, the 1992-2012 trend was calculated as (FN2012 – FN1992)/FN1992*100. See Hirsch et al. (2010), Hirsch et al. (2018a) and Choquette et al. (2019) for a complete description of the trend methods, including the weighted regression approach and the flow-normalization process. Furthermore, to gauge uncertainty of the trends, likelihood estimates of the trend direction for each site and parameter were extracted from Murphy et al. (2018). These estimates are described in Oelsner et al. (2017) and Murphy et al. (2018) and use

a block-bootstrapping approach presented in Hirsch et al. (2015) and Hirsch et al. (2018b). Upward or downward trends were considered "likely" if the likelihood was > 0.85, "somewhat likely" if the likelihood was from 0.85 to 0.70, and "as likely as not" to be upward or downward if the likelihood was < 0.70. The efficacy of using WRTDS for estimating trends in sediment concentration and flux has been explored and discussed in Moyer et al. (2012), Chanat et al. (2016) and Lee et al. (2016).



## 2.2 Description of watershed data and changes

For each site, 52 estimates of land use/cover, hydro-climatic change and static/long-term watershed characteristics (Table 1) were generated or compiled using data from Falcone (2017) and Farmer et al. (2017). Falcone (2017) includes time-series variables characterizing land use/cover and climate for each watershed and for a near-site near-stream zone, which was

computed as 25% of the watershed area nearest the site and stream (Fig. 1) and hereafter referred to as the proximal zone. For this study, the percent change, relative to the starting condition, of each variable was calculated using the years closest to 1992 and 2012, i.e. (variable2012 – variable1992)/variable1992*100. The spatial resolution and frequency of data collection varied by variable. Data collected at the annual time scale were smoothed using locally weighted regression (loess) prior to calculating percent change to characterize the systematic change in these variables over time. The static/long-term watershed

characteristics were also extracted from Falcone (2017). Three variables characterizing trends in streamflow were retrieved from Farmer et al. (2017) for each site. See Table 1 for a list of all variables and brief descriptions. See Falcone (2017) and Farmer et al. (2017) for specific information about data processing and original source information for these data.

Each site was also assigned 1 of 4 categories describing the predominant land use in the corresponding watershed (urban,

agricultural, undeveloped or mixed-use), based on the categorization scheme provided in Falcone (2015). See Supplemental Material for the explicit land-use categorization scheme used in this study. Across all 137 sites with either SSC or TSS data, 7 sites switched land-use categories between 1992 and 2012. All seven watersheds became more urban, shifting categories from mixed-use, agricultural or undeveloped to urban or mixed-use. For consistency, and because the specific reason(s) that caused a site to switch land-use categories presumably corresponds to the land-use category at the end of the record (e.g.

increased urbanization caused an agricultural watershed to become an urban watershed), the 2012 land-use categorization was used to group sites.

## 2.3 Methods for exploring potential drivers of change

Since the potential drivers of systematic, multi-decadal changes in sediment in US rivers and streams are varied, one useful

approach is to conceptualize changes in sediment as a function of changes in the streamflow regime versus changes on the landscape (Choquette et al., 2019; Murphy and Sprague, 2019). Thus, each sediment trend was parsed into two components of change: the amount of change due to changes in the streamflow regime, i.e. the streamflow trend component (QTC), versus the amount of change in sediment due to changes in landscape management, i.e. the management trend component (MTC). These estimates, which can be found in Murphy et al. (2018), were compared across watershed land uses,

geographic location and sediment trend magnitudes. Choquette et al. (2019), Hirsch et al. (2018a) and Murphy and Sprague (2019) provide details about the method used to parse water-quality trends into QTC and MTC contributions.



Briefly, the sediment trend can be described as an additive function of the MTC and QTC, the absolute and relative contributions of which provide insight into broad drivers of change (Choquette et al., 2019; Murphy et al., 2019). MTC is estimated as the sediment trend assuming a stationary streamflow regime. As such, MTC describes the potential amount of change in sediment concentrations over time due to factors other than long-term systematic changes in the streamflow

regime. This estimate isolates the amount of change in sediment due to changes in the concentration-streamflow (C-Q) relationship (Choquette et al., 2019; Murphy and Sprague, 2019), also often referred to as a sediment rating curve. Changes in C-Q relationships are often used to identify and understand human influences on water quality (e.g. Moatar et al., 2017; Murphy et al., 2014; Basu et al., 2010; Bieroza et al., 2018). Choquette et al. (2019) and Hirsch et al. (2018a) refer to the MTC as the CQTC (concentration-streamflow trend component) but this analysis uses the more conceptual terminology

presented by Murphy and Sprague (2019). Analytically, the MTC is estimated using WRTDS and a stationary streamflow regime is specified during the flow-normalization procedure. When the MTC is subtracted from the sediment trend, this gives the QTC. QTC describes the potential amount of change in sediment concentrations over time due specifically to long-term, sustained changes in any aspect of the streamflow regime. These could be changes in the magnitude, timing or frequency of streamflow that ultimately effect sediment. Taken together, the sediment trend is the sum of the MTC and QTC.

See Choquette et al. (2019), Hirsch et al. (2018a) and Murphy and Sprague (2019), for a complete description of these methods including example applications at individual sites and extended discussion on interpreting these types of estimates. Additionally, while Murphy and Sprague (2019) present estimates of MTC and QTC for sediment at some of the same sites in this paper (though for a longer trend period), the results presented here greatly expand that initial investigation by comparing these estimates regionally, by land use, to the magnitude of the overall change in sediment, and to observed

hydro-climatic changes.

Finally, to link sediment trends to specific changes in land use/cover, hydrology and climate, plus static/long-term watershed characteristics that might influence how responsive a site is to change, a correlation analysis was completed. The Kendall's Tau correlation coefficient (Kendall, 1938) was computed between the sediment trend, in percent change, and each of the 34

potential causal variables (13 land use/cover of the watershed and 12 of the proximal area, 9 hydro-climatic changes, Table 1). For the 18 static/long-term watershed characteristics (Table 1), the sediment trend in absolute percent change was used. These potential causal variables were selected because they characterize, or partially characterize, some possible driver of sediment concentration and the available data were temporally and spatially consistent. The static/long-term watershed characteristics describe various physical features of the watershed that could influence the sensitivity of sediment at a site to

changes in the watershed.  Kendall's Tau is a non-parametric alternative to Pearson's correlation and was used because of the non-normal distributions of the variables. All analyses were completed using the R statistical software program (R Core Team, 2018).



## 3.0 Results and Discussion

### 3.1 Sediment concentration trends

Since 1992, sediment concentrations largely decreased at the 137 rivers and stream sites across the contiguous US. These downward trends were more common and had larger magnitudes for SSC trends compared to TSS trends (Table 2). Larger
percent decreases tended to occur at sites with high concentrations in 1992 whereas the largest percent increases occurred at sites with low starting concentrations (Fig. SM-1). Starting concentrations were typically lower for TSS compared to SSC (Table 2) reflecting a combination of differences in analytical procedures and different sets of sites.

Sediment trends varied geographically across the US. Between 1992 and 2012, increases in SSC occurred exclusively at sites
in the eastern US (Fig. 2), and median percent changes in SSC were -45%, -23% and 5% for sites in the western, central and eastern US, respectively (Fig. 3a). TSS trends mostly decreased at sites in the central US, with a median percent change of about -23% (Fig. 3a). However, TSS trends had the opposite pattern of change for sites in the western and eastern US compared to SSC trends; median percent changes for TSS trends were 18% and -17%, respectively (Fig. 3a). The location of sites within each geographic region differs between parameters. For example, in the western US, sites with TSS trends were
clustered in the northwestern US while sites with SSC trends were spread across the western US more generally (Fig. 2). Similarly, in the eastern US, sites with TSS data were spread across the region while there were almost no sites with SSC trends in the southeastern US (Fig. 2). These differences in the geographic distribution of SSC compared to the TSS sites, at least partially, account for some of the differences in trend direction and magnitude between sediment parameters in the same geographic region.

Like other studies, land-use categorization of the watershed yielded different patterns of sediment trends (Oelsner and Stets, 2019; Lacher et al., 2019). At agricultural sites, there was an overall decrease between 1992 and 2012 for both SSC and TSS. At urban, undeveloped and mixed-use sites, the patterns of trends differed between sediment parameters. For SSC, undeveloped sites had the largest decreases with a median percent change of -41%. Urban sites and mixed-use sites had a
larger proportion of upward SSC trends with median percent changes of 3% and -6%, respectively (Fig. 3b). For TSS, undeveloped sites had the largest proportion of upward trends and some of the largest increases in TSS compared to sites in other land-use categories. Urban sites and mixed-use sites typically had decreases in TSS, with median percent changes of -13% and -21%, respectively (Fig. 3b). The proportion of sites with various watershed land uses is relatively similar across geographic regions for both sediment parameters. For example, undeveloped sites largely occur in the western US and
agricultural sites largely occur in the central US for both sediment parameters (Fig. SM-2). Thus, the stark difference between the largely downward SSC trends and largely upward TSS trends at undeveloped sites in the western US could be due to differences in the causes of the changes for undeveloped sites in the northwestern US compared to other undeveloped

sites in the western US (Fig. 2). Other contributing factors could include differences in the suspended particle-size distributions being characterized by SSC and TSS and different regions having different underlying geology. For example, the underlying volcanic and metamorphic geology in the northwestern US, coupled with the steep terrain, likely results in larger particle sizes and less accurate TSS estimates compared TSS estimates from regions underlain by sedimentary rocks,

having predominately finer sediment particle sizes in streams.

## 3.2 Land management changes

Murphy and Sprague (2019) showed that MTC is typically the dominant contributor to trends in concentration for sediment and other water-quality parameters such as nutrients, major ions and salinity. This study, which uses a shorter trend period

than Murphy and Sprague (2019), found 83% of the sites had larger absolute values of MTC than QTC (Table 3). This pattern held across all land-use categories and most sites (Fig. SM-3, Table 3), indicating changes in land management typically had a greater influence on sediment transport than changes in the streamflow regime alone. Furthermore, MTCs tended to be negative, mirroring the overall sediment trend (Fig. 4a). About 80% of the SSC trends and 60% of the TSS trends had negative MTCs, suggesting that, at most sites, management actions on the landscape likely led to decreases in

sediment concentration (Table 3).

Changes in land use/cover are often proposed as a major driver of changes in sediment over time. Studies that support this hypothesis have used methods such as modelling (Crossman et al., 2013; Naik and Jay, 2011; Nelson and Booth, 2002; Lacher et al., 2019), parsing of water-quality and streamflow records (Li et al., 2016; Wu et al., 2012; Shen et al., 2017;

Murphy and Sprague, 2019), conducting small field- or catchment-based studies (Bartley et al., 2010; Vogl and Lopes, 2010) or using general qualitative statements with and without quantitative land-use information (Kreiling and Houser, 2016; Gitau et al., 2010; Panthi et al., 2017). The MTC estimates provided here are similar in magnitude and direction of the SSC and TSS trends for many of the sites (Fig. 4a), providing additional support for this hypothesis.

Sediment trends were moderately to strongly correlated (abs(Tau) > 0.4) with several specific changes in land-use/cover, depending on the predominant land use of the watershed (Fig. 5). For the most part, SSC trends were well correlated with changes in land use/cover in watersheds that had more anthropogenic and homogenous land uses. For example, SSC trends at urban sites were well correlated with 8 variables indicative of urbanization and at agricultural sites SSC trends were well correlated with 10 variables characterizing changes in agriculture or moderate development (Fig. 5). Undeveloped and

mixed-use sites were well correlated with fewer land-use/cover change variables across the watershed and proximal zone. Note, many of the moderate to strong correlations between potential causal variables and SSC trends were not statistically significant at the 0.05 level due to a variety of reasons, one of which is likely the small number of sites in some of the land-





use categories. In general, TSS trends were not well correlated with many of the 13 watershed or 12 proximal land-use change variables. One exception was the TSS trend in urban sites that were well correlated with 5 variables describing a mix of urban and agricultural land-use/cover changes in the watershed and proximal zone (Fig. 5). The lack of well-correlated variables may be due more to the uncertainty introduced during TSS determinations than to a true lack of potential causes for

changes in sediment at these sites. Though, TSS trends characterize changes in smaller sediment sizes (i.e. silts and clays) compared to SSC, and perhaps correlating changes in the concentration of smaller particles with land-use/cover changes is more challenging than somewhat larger, sand-sized sediment, making the correlations between TSS and these variables weak. For example, changes in landscape management may disproportionally influence a particular size of sediment. Some BMPs, such as retention ponds and check dams in small channels, slow streamflow velocities, which in turn encourages

larger-sized sediment to drop from suspension. These types of BMPs may have little effect on small, silt- and clay-sized particles. This may be part of the reason TSS trends were not well correlated with many of the land-use/cover variables. Other types of BMPs, such as grass filters, settling ponds and grassed waterways, tend to trap silt and the effect of such installations would likely be captured by TSS and SSC determinations of sediment concentration. Thus, information about BMPs, including type, installation dates, and the density of installations across a watershed provide important information

for characterizing their possible effect on sediment nearby or downstream. However, this information is often difficult to obtain and aggregate.

Several studies have compared water quality to land-use/cover characterizations based on the entire watershed and to a more confined riparian or buffer area. For example, Johnson et al. (1997) found that buffer-zone characterizations of land

use/cover were a better indicator of water quality compared to land use/cover across the watershed; whereas, Sliva and William (2001) and Hunsaker and Levine (1995) found the opposite. In this current study, explicit characterization of land-use/cover changes in the proximal zone typically did not yield more or stronger correlations with sediment trends compared to land-use/cover changes across the entire watershed (Fig. 5). For SSC, more trends were well correlated with land-use/cover changes across the watershed (16 variables) compared to just the proximal zone (10 variables). For TSS, 2 of the

watershed variables, compared to 4 of the proximal zone variables were well correlated with TSS trends, and 7 watershed variables compared to 6 proximal zone variables were statistically significant (alpha = 0.05; Fig. 5). Various studies have speculated why riparian/buffer zone characterizations do not necessarily provide more explanatory power given the known importance of riparian zone and near-stream conditions on water quality at a local scale (Hunsaker and Levine, 1995; Johnson et al., 1997; Sliva and William, 2001). In this study, most of the proximal zone land-use change variables that were

correlated with the sediment trend were similar to the watershed-based estimates, however, there were some exceptions. For example, the percent change in the percent of agricultural land (row crops and pasture) enrolled in CRP in the proximal zone was well correlated with changes in SSC at agricultural sites, whereas the percent change across the whole watershed was not (Fig. 5).





Some land-use/cover changes correlated with sediment trends in a counterintuitive direction. For example, SSC trends at agricultural and undeveloped sites were positively correlated with the percent change in agricultural land in the proximal zone enrolled in CRP (Fig. 5). Only 4 agricultural sites had an increase in CRP and only 1 of these 4 had a corresponding

increase in SSC (Fig. 6a). This SSC trend was a small <10%, "as likely as not", increase related to a 200% increase in CRP, suggesting the increase in CRP at this site had little influence on sediment concentration. One undeveloped site had an increase in CRP that was related to an ~25% decrease in SSC (Fig. 6a).  Thus, these positive correlations provide little information about the influence of changes in CRP on sediment concentrations in streams because the results are largely influenced by data from a few sites. Interestingly, while the correlation between TSS trends and the CRP change variables

("Ag land in CRP" and "Watershed in CRP" for the entire watershed and the proximal zone) are very weak or slightly negative (Fig. 5), the relationship of percent change in agricultural land in the proximal zone enrolled in CRP ("Ag land in CRP") to TSS trends is likely a more accurate reflection of the effects of CRP on sediment concentrations in rivers (Fig. 6b).

Many agricultural and non-agricultural sites with TSS data showed increases in CRP and most were associated with

decreases in TSS (Fig. 6b). This provides evidence, though limited, that increases in the proportion of the agricultural land in the proximal zone enrolled in CRP may lead to decreases in sediment concentration. Two sites had a substantial percent increase in TSS associated with a very large percent increase in CRP (Fig. 6b). Both sites are undeveloped watersheds in the northwestern US and have some of the lowest sediment concentrations in the dataset (e.g. between 3-7 mg/L in 1992) and increased by only 2 or 3 mg/L.  Decreases in sediment concentration also appear to be related to how much land in the

watershed was enrolled in CRP at the beginning of the trend period. When more than 2% of a watershed was enrolled in CRP in 1992, sediment almost always decreased by 2012 (Fig. 6c and d), which suggests several potential causes including long lag times between vegetation/soil health improvements and water-quality recovery, that 1992 CRP enrollment represented a commitment to better farming practices, or that enrollment in CRP across the watershed needs to be above a certain threshold for effects of these changes to be seen as changes in riverine sediment transport. Not surprisingly, this

relationship was most obvious for agricultural sites, since urban, mixed-use and undeveloped sites likely have other land-use/cover or land-management changes driving changes in sediment concentration.

Some of the largest correlations occurred between sediment trends and changes in urbanization land-use/cover variables at urban sites (Fig. 5). Increases in low-medium density dwellings likely indicate additional or new construction and

earthmoving activities on previously less developed lands. This construction disrupts the landscape, increases erosion and often leads to degraded water quality as larger quantities of sediment are transported to the stream. The MTC and the SSC trend both have a similar relationship to changes in low-medium density dwellings whereas QTC does not (Fig. 7). This finding suggests that changes in the number of low-medium density dwellings may have led to large shifts in the C-Q





relationship (the MTC) but had little to no effect on the streamflow regime. Interestingly, despite continued urbanization and increases in some land uses often associated with worsening water quality, sediment concentration still largely decreased at urban sites (Fig. 3b). For example, increases in SSC only occurred at sites where low-medium density dwellings increased by 30% or more (Fig. 7). At other sites, small and moderate increases in low-medium density dwellings occurred at sites

with decreases or little change in SSC, suggesting conservation efforts to reduce sediment runoff to streams may be successful, up to a point, even as lands continue to be urbanized.

### 3.3 Hydro-climatic changes

The QTC provides a general estimate of the amount of change in sediment due exclusively to changes in the streamflow

regime. When the QTC is large, in absolute terms, natural or human activities could be causing these changes. For example, systematic changes in climate due to increased greenhouse gases in the atmosphere, quasi-periodic fluctuations in climate (such as the El Niño-Southern Oscillation), changes in dam operations or extensive alteration of the stream channel (e.g. straightening or channelization), could all induce a change in streamflow over time, which in turn could lead to changes in transport, resuspension and erosion of sediment within the channel, riparian zone and floodplain. Using a slightly longer

trend period, Murphy and Sprague (2019) found sediment trends, compared to other water-quality parameters, were more likely to be comprised of contributions of both MTC and QTC. Similarly, this study finds around 60% of the sites had non-negligible QTC contributions (> +/- 5% change) to sediment trends. These contributions tend to be much smaller than MTCs (Fig. 4a) and only about 17% of the sites had a QTC that exceeded the MTC (Table 3).  At a limited number of sites, changes in streamflow accounted for almost the entire change in sediment (10% of SSC sites and 20% of TSS sites; Table 3), though

many of these sediment trends were small. Only 1 of the SSC trends and 10 of the TSS trends had both an increase (or decrease) of at least 5% that was almost entirely due to changes in the streamflow regime with little to no contributions from changes in land management. Thus, while changes in the streamflow regime were typically not the dominant driver of changes in sediment concentration, they were often a contributing influence and at a few sites were the main driver of change (Fig. SM-3).

The correlative strength of hydro-climatic changes with sediment trends across land-use categories and parameters was not uniform (Fig. 5). None of the hydro-climatic variables were correlated with sediment trends at undeveloped sites for either parameter suggesting that climate change and climate variability alone were not sufficiently strong to affect sediment concentrations across these sites. There is limited consensus on how changes in climate thus far have influenced sediment in

rivers (see references in Whitehead et al. (2009) and Wohl (2015)). Previous models have suggested that changes in climate will lead to increases and decreases in sediment in particular rivers or areas of the western US (Records et al., 2014; Ficklin et al., 2013). However, human influences, especially dam construction and management, have been shown to be important



drivers of change in other areas (Walling, 2009; Rossi et al., 2009; Williams and Wolman, 1984). Surprisingly, the results in this study suggest a limited influence from dams on sediment trends. Changes in the storage capacity of major dams and changes in the number of dams in the watershed were only well correlated with sediment trends at urban sites (Fig. 5) -- only 2 sites showed a change in either variable, both of which had small (~7%) or moderate (~25%) increases in sediment.

However, neither site was close to a dam (both >5 km downstream of a dam) and these increases in the number of dams and dam storage may be occurring much farther upstream from the site. In fact, only 7 and 10 sites with SSC and TSS data, respectively, were within 5 km downstream of a dam. Also, the direction of the sediment trends at these handful of sites were mixed, and across all sites the number of dams and dam storage volume increased only between 1992 and 2012 (i.e. no site in this dataset had a decrease in the number of dams or amount of storage in the watershed). Thus, the limited effect of dams

on sediment trends that was observed in this study is likely because the characteristics of this dataset and the included sites are not optimal for exploring the effect of dams in detail. Additional work explicitly considering sites closer to dams and information such as dam proximity, the proportion of total streamflow controlled by dams and trapping efficiency of upstream dams would further illuminate this potential driver of change.

Variables characterizing trends in specific annual metrics of daily streamflow ("Q Slope: mean day", "Q Slope: max day", and "Q Slope: 7-day min") had a few moderate correlations with sediment trends (Fig. 5). The QTC estimates indicate stronger influences from changes in the streamflow regime on sediment trends than is apparent from the correlation analysis (Table 3). It is possible the hydro-climatic change variables associated with streamflow were not strongly correlated with sediment trends because changes in the streamflow regime were masked or offset by larger coincident changes in land

management. This effect was seen for other water-quality parameters as well in Murphy and Sprague (2019). Figure 7 demonstrates this effect by comparing bivariate plots of the sediment trend, MTC and QTC to a land-use change variable and hydro-climatic change variable. The sediment trend is not well correlated with a change in annual mean daily streamflow. However, when changes in mean daily streamflow are compared to just the QTC there is a well-defined positive relationship indicating that large decreases in streamflow relate to large decreases in sediment concentration and the same for positive or

increasing changes (Fig. 7). This pattern is quite different from when the percent change in low-medium density dwellings is compared to the sediment trend, MTC and QTC. These relationships suggest that decreases in streamflow decrease sediment transport or resuspension in the stream, but these improvements are partially offset by human activities in the watershed, such as increases in low-medium density dwellings. Thus, the lower correlations may be downplaying the importance of changes in the streamflow regime on sediment trends.

An additional consideration is that changes in streamflow can also induce a change in the C-Q relationship, and this response may be more common for sediment compared to other water-quality parameters. Recall the MTC captures the influence of changes in the C-Q relationship on sediment concentration, thus if the streamflow regime changed in such a way to perturb



the C-Q relationship this effect would be captured by the MTC. C-Q relationships have been shown to vary by storm depending on a host of hydrologic and antecedent conditions, and over short time periods due to droughts or highly wet years (Duncan et al., 2017; Biron et al., 1999). However, sustained, systematic changes in the C-Q relationship due exclusively to changes in the streamflow regime are less well documented (e.g. Bieroza et al., 2018). A few of the hydro-

climatic change variables were well correlated with MTC, though again, only at urban and agricultural sites (Fig. 8). This finding suggests the limited ability of hydro-climatic changes to shift the C-Q relationship over time, at least for sediment concentrations at these sites. QTC was much more strongly correlated with hydro-climatic change variables across all land-use categories and for both sediment-sample types (Fig. 8), showing the importance of changes in streamflow on sediment, even if these changes are often masked or counteracted. Understanding the sensitivity of the C-Q relationship (i.e. the MTC)

to systematic changes in the streamflow regime would further illuminate the effects of such changes on sediment concentration in rivers and streams.

### 3.4 Importance of location

Sediment dynamics are strongly influenced by geographic location and are particularly sensitive not only to land use of the

watershed, floodplain and riparian zone but also the geologic, pedologic, climatologic, physiologic, hydrologic and geomorphologic conditions of the site (Charlton, 2007). The location of a sampled site in a fluvial system's longitudinal profile can be an important factor in the types and amounts of sediment available for transport particularly if that stretch of river is supply-limited or transport-limited. Similarly, channel evolution processes are an important determinant of sediment dynamics.  For example, if a site is located on or downstream of a length of river that shifts from an aggradation to

degradation phase this would change sediment concentrations over time. Gellis et al. (1991) found that decreases in sediment and salt loads in the Colorado River basin were likely due to a natural shift in incised-channel evolution, which includes sequential phases involving channel deepening, widening and then deposition of a floodplain. Changes in sediment loads related to this natural geomorphic process were further exacerbated by concurrent changes in the streamflow regime (Gellis et al., 1991). These natural factors can influence not only a site's capacity for change but also its recovery potential

(Charlton, 2007). Multiple static and long-term watershed conditions (Table 1) were used to explore the sensitivity of sediment trends to location. Surprisingly, only a few of the land-use categories had sediment trends that were well correlated with one or more of the 18 static/long-term watershed characteristics, again more so for SSC trends than TSS trends (Fig. 5). Since the sediment trends were in terms of absolute change for these correlations, positive (negative) correlations indicate increased (decreased) sensitivity of sediment concentrations at a site as the gradient of a given static/long-term watershed

characteristic increases, leading to larger (smaller) sediment trends. For example, SSC trends at undeveloped sites were negatively correlated with long-term relative humidity, indicating smaller changes in sediment (increases or decreases) occurred at sites with higher relative humidity. Sites with high relative humidity tend to also be more vegetated and the





amount of sediment readily available for transport at these sites is less than at a more arid site. Thus, SSC at more arid sites is more sensitive to hydrologic or land-management changes than at humid sites.

The MTC and QTC estimates suggest location is important for understanding the potential drivers of change in sediment

concentration. When grouped broadly by geographic region, western US sites, which also are often undeveloped (Fig. SM-2), typically have negative QTCs for both SSC and TSS trends, indicating changes in streamflow tend to lead to decreases in sediment in this region (Fig. 4b and Fig. 4c). Furthermore, western US sites also appear to cluster according to whether the QTC and MTC have the same or opposing signs, which indicates whether streamflow enhanced or offset the effects of concurrent changes in land management on sediment. For decreasing SSC trends at western sites, negative MTCs coupled

with somewhat smaller but also negative QTCs suggest changes in the streamflow regime further enhanced larger decreases in sediment from changes in land management (Fig. 4b).  For increasing TSS trends at western sites, positive MTCs coupled with somewhat smaller negative QTCs suggest the opposite; changes in streamflow partially offset potential increases in sediment due to changes in land management (Fig. 4c). This effect results in somewhat smaller sediment trends than would have been observed if the streamflow regime had remained stationary over this period. Central and eastern US sites show a

mix of opposing and reinforcing effects of the MTC and QTC on sediment trends (Fig. 4b and Fig. 4c). For both parameters, roughly about half the sites had opposing effects and the other half had reinforcing effects on the sediment trend (Table 3; Fig. SM-3). However, it was relatively rare for both the QTC and MTC to be positive, indicating it was uncommon for increases in sediment to be due to increases from both changes in land management and changes in streamflow. Instead, increases from changes in land management were more often slightly offset by decreases from changes in the streamflow

regime (negative QTCs; Fig. 4b and Fig. 4c). Finally, the QTC was near zero for about half of TSS trend sites (Table 3), largely in the eastern US (Fig. 4c), unlike SSC trends where fewer sites had QTCs near zero (Fig. 4b). Since TSS trends tend to capture changes in a finer suspended particle-size distribution than SSC, it may be that TSS trends are less sensitive to changes in the streamflow regime, particularly when these changes occur at higher streamflow, which has been observed in many rivers in the eastern US (e.g. Armstrong et al., 2014). Compared to the QTC, it was much less common for the MTC to

be near zero, especially when changes in SSC or TSS were moderate to large in magnitude (Fig. 4b and Fig. 4c). Only 1 SSC trend and 10 TSS trends had a change in sediment that was greater than +/- 5% and the MTC was near zero, indicating that at select sites, changes in sediment were almost exclusively due to changes in the streamflow regime (Table 3).

### 3.5 Limitations

In many ways, the datasets used here provide a greater breadth and depth of information for exploring potential causes of sediment trends compared to previous studies: the datasets are temporally consistent thus comparable over time; spatially consistent allowing for comparison across sites; publicly available with well-documented metadata; and spatially explicit



allowing for estimates of the entire watershed or the proximal zone (Falcone, 2017). However, even with this extensive information, it was difficult to identify specific potential causes of sediment trends for some land-use categories. Additionally, because multiple correlations were completed (208 for each set of SSC and TSS trends shown in Fig. 5), about 2 and 10 of the statistically significant correlations at the alpha 0.01 and 0.05 level, respectively, can be expected to be false positives.

The difficulty of establishing clear, straightforward relationships between potential causal variables and sediment trends presents a real challenge for researchers, especially those working with streams across a large geographic region. It is possible the choice of potential causal variables used in these analyses did not capture the relevant changes at these sites. Other variables, if available, may better characterize important changes on the landscape or to watershed management. There is quite possibly a disconnect between the conceptual "land-management changes" identified while parsing the sediment trends and the land-use/cover change information that was available for the correlation analysis. Information that could be helpful but often is not available in a nationally or temporally consistent dataset include channel and floodplain geomorphological characteristics, construction activities near the site, types and density of riparian vegetation, BMP information, changes in dam operations and if the site is undergoing channel evolution and what phase the site is in (i.e. degradation or aggradation).

Additionally, the use of TSS to characterize changes in sediment may make exploring potential causes more difficult than when SSC is used. As described in Section 2.1, TSS tends to measure the concentration of smaller suspended particle sizes as opposed to the entire suspended particle-size distribution of a sample and has been "shown to be fundamentally unreliable for the analysis of natural-water samples" (Gray, et al., 2000). The uncertainty inherent in TSS estimates is a likely explanation for the lack of correlation between TSS trends and potential causal variables (Fig. 5). However, heterogeneity across the sites and difficulty in identifying potential causes of change in only the fine suspended-sediment fraction (less sands) are other possibilities. TSS and SSC trends can also give quite different results. In Murphy et al. (2018), 5 sites with 2002-2012 trends and 3 sites with 1992-2012 trends had both types of sediment data. These trends had different magnitudes, and a few had different trend directions depending on the sediment parameter (Fig. SM-4), which supports the conclusion of the incompatibility of SSC and TSS estimates as shown by Gray et al. (2000). Thus, caution should be taken when comparing TSS and SCC results and researchers should note the possible difficulties when using TSS estimates to understand changes in sediment and their potential drivers.

Finally, sediment may present a relatively unique challenge when trying to identify potential causes of trends compared to other water-quality parameters, such as pesticides or nutrients. Sediment transport is fundamentally different from other water-quality parameters, relying on the physical properties of fluid dynamics as opposed to chemical reactions. For





example, streambank erosion can be a dominant contributor of suspended material to a river, and while adjacent land use/cover and management can be important in determining the amount of erosion (Fox et al., 2016), it is also possible that channel erosion is more strongly related to channel properties and conditions, such as channel roughness, slope, sinuosity and near-channel vegetation density and type (Charlton, 2007). Changes in these variables are difficult to track over time and

a unified dataset containing such information for multiple sites across a specific geographic region or the US is non-existent. Also, the source and mobilization of sediment can be natural or human-influenced and includes the remobilization of legacy sediment (Wohl, 2015). Changes in land management may have led to the deposition of sediment stores in the channel and floodplain, but a change in the streamflow regime may be the ultimate factor causing the erosion and transport of the stored sediment downstream. Lastly, the causes of changes in sediment transport vary based on the time scale. For example,

changes occurring within a single decade are more strongly related to weather patterns compared to changes occurring over centuries, which are more strongly related to tectonics (Vercruysse et al., 2017; Charlton, 2007).

## 4.0 Conclusion

Annual mean concentrations of suspended sediment largely decreased between 1992 and 2012 at 137 stream sites with

watershed areas < 300,000 sq km across the contiguous US. Many of these decreases occurred at sites with some of the highest concentrations and at sites that drained watersheds with concurrent small to moderate increases in human-related land uses (i.e. urban and agricultural land uses), suggesting efforts to minimize sediment pollution to streams and rivers may be having the desired effect in some places. At many locations, a change in land management (including changes in land use/cover), as opposed to a change in the streamflow regime, was the primary contributor of changes in sediment, though

systematic changes in the streamflow regime often had a mild-to-moderate influence on sediment. The influence of specific hydro-climatic changes on sediment trends appears to be masked due to more influential changes in land management. Surprisingly, characterizing land-use/cover changes in the proximal zone provided little additional information compared to characterizing land-use/cover changes across the whole watershed, indicating a need for more precise information about near-channel conditions. Sediment trends determined using TSS data were weakly correlated with potential causal variables,

highlighting the difficultly of using TSS, as opposed to SSC, data to infer potential causal relationships largely due to the unreliability of TSS for characterizing stream water quality. While identifying the specific land use/cover or hydro-climate changes responsible for these sediment decreases remains a challenge, the strongest correlations tended to occur at sites with more homogenous, human-related land uses (i.e. agricultural and urban lands). At many sites, across all land-use categories, decreases in sediment are likely due to changes in land management with changes in the streamflow regime providing a

limited though important and often overlooked influence.





**Data availability**

Site information and the sediment concentration and streamflow data used to estimate the trends are published at http://dx.doi.org/10.5066/F7KW5D4H (De Cicco et al., 2017), the land use data are available at https://doi.org/10.5066/F7TX3CKP (Falcone, 2017), the streamflow trend data are available at http://www.dx.doi.org/10.5066/F7D798JN (Farmer et al., 2017), and the estimates of the sediment trends, management trend components (MTC) and streamflow trend components (QTC) are available at https://doi.org/10.5066/F7TQ5ZS3 (Murphy et al., 2018).

**Author contribution**

Jennifer Murphy was the sole contributing author and completed the data processing and analysis and report writing.

**Competing interests**

I have no competing interests.

**Acknowledgements**

This work would not have been possible without the nationwide trends assessment completed by the USGS National Water Quality Assessment Project's Surface Water Status and Trends team. Special thanks to Lori Sprague, Gretchen Oelsner, Henry Johnson, Edward Stets, Melissa Riskin and Karen Ryberg for compiling, processing, screening, and harmonizing the interagency data, reviewing model diagnostics, and performing auxiliary tasks related to this effort. Additional, thanks to James Falcone for compiling and synthesizing the land-use change, climatic change, and static/long-term watershed variables used in this study. Finally, thank you to the many hydrologist and hydrologic technicians who collected these streamflow and water-quality data year after year; without that sustained commitment, this work would not have been possible. Any use of trade, firm, or product names is for descriptive purposes only and does not imply endorsement by the U.S. Government.



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

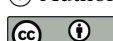



**Table 1. Potential causal variables and static/long-term watershed characteristics used in correlation analyses. Streamflow trend variables ("Q slope: mean day", "Q slope: max day", and "Q slope: 7-day min") are from Farmer et al. (2017); all other variables published in Falcone (2017), see publications for details and original source information. ## and #### symbols in variable names indicate 2-digit or 4-digit year of data value. Percent-change computations are (trend end-year value - trend start-year value) / trend start-year value * 100. [square kilometer, sq km; meters, m; centimeters, cm; degrees Celscius, °C]**

| Short name | Data description (original time-series or static variable name from referenced source) |
|---|---|
| **Land-use and land-cover changes across entire watershed or proximal zone[1]** | |
| Agricultural land | Percent change in agricultural land, excluding potential grazing lands, as a percentage of the watershed (NWALT##_AG4344_SUM) or proximal zone (RIP_NRSITE_NWALT##_AG4344_SUM) |
| Ag+Grazing land | Percent change in agricultural land, including potential grazing lands, as a percentage of the watershed (NWALT##_AG4346_SUM) or proximal zone (RIP_NRSITE_NWALT##_AG4346_SUM) |
| Cropped land | Percent change in row-cropped land as a percentage of the watershed (NWALT##_43) or proximal zone (RIP_NRSITE_NWALT##_43) |
| Ag land in CRP | Percent change in proportion of agricultural land enrolled in the Conservation Reserve Program (CRP) across the entire watershed (crp.crop##) or proximal zone[2] (RIP_NRSITE_CRP_CROP##) |
| Watershed in CRP | Percent change in amount of land enrolled in CRP as a percentage of the watershed (NWALT##_AG4344_SUM * crp.crop## * 0.01) or proximal zone (RIP_NRSITE_NWALT##_AG4344_SUM * RIP_NRSITE_CRP_CROP## * 0.01) |
| All developed land | Percent change in developed and semi-developed land as a percentage of the watershed (NWALT##_DEV_SUM + NWALT##_SEMIDEV_SUM) or proximal zone (RIP_NRSITE_NWALT##_DEV_SUM + RIP_NRSITE_NWALT##_SEMIDEV_SUM) |
| Developed land | Percent change in developed land as a percentage of the watershed (NWALT##_DEV_SUM) or proximal zone (RIP_NRSITE_NWALT##_DEV_SUM) |
| Semi-developed land | Percent change in semi-developed land (land in close proximity to developed lands and partially used for same purposes) as a percentage of the watershed (NWALT##_SEMIDEV_SUM) or proximal zone (RIP_NRSITE_NWALT##_SEMIDEV_SUM) |
| Impervious area | Percent change in impervious land cover as a percentage of the watershed (NWALT_IMPV##) or proximal zone[2] (RIP_NRSITE_NWALT_IMPV##) |
| Low-med density dwellings | Percent change in land with low-medium density residential development as a percentage of the watershed (NWALT##_26) or proximal zone (RIP_NRSITE_NWALT##_26) |
| Low-use land | Percent change in land with little to no development or agriculture as a percentage of the watershed (NWALT##_50 + NWALT##_60) or proximal zone (RIP_NRSITE_NWALT##_50 + RIP_NRSITE_NWALT##_60) |
| Mining & related activities | Percent change in mined land as a percentage of the watershed (NWALT##_41) or proximal zone (RIP_NRSITE_NWALT##_41) |
| Fracking wells | Percent change in the mean number of hydraulic fracturing wells within subwatersheds as a percentage of the watershed (frac_wells####) |
| **Hydro-climatic changes** | |
| Total precip | Percent change in total precipitation across watershed (sum of monthly mean precipitation, see Falcone (2018)) |
| Average temp | Percent change in annual mean monthly temperature across watershed (mean of monthly mean temperature, see Falcone (2018)) |
| Temp range | Percent change in monthly mean temperature range across watershed (difference between hottest and coldest monthly mean temperature of the same year, see Falcone (2018)) |
| Max temp | Percent change in annual maximum monthly temperature across watershed (maximum of monthly mean temperature, see Falcone (2018)) |
| Q slope: mean day | Slope of annual mean daily streamflow trend as percent change per year ((e^meanL_slope - 1)*100) |
| Q slope: max day | Slope of annual maximum daily streamflow trend as percent change per year ((e^maxL_slope - 1)*100) |
| Q slope: 7-day min | Slope of annual 7-day minimum streamflow trend as percent change per year ((e^min7nL_slope - 1)*100) |
| Density of major dams | Percent change in the number of major dams per 100 sq km across watershed (MAJDAMS_100sqkm_####) |
| Dam storage | Percent change in dam storage per sq km across watershed (NORMSTOR_sqkm_####) |
| **Static/long-term watershed characteristics** | |
| Drainage area | Drainage area of watershed, in sq km (gisareakm2) |
| Basin compactness | Watershed compactness ratio (area/perimeter^2 * 100), higher number means more compact (circular) shape, unitless (bas_compactness) |
| Average elevation | Mean watershed elevation, in m (ELEV_SITE_M) |
| Stdev of elevation | Standard deviation of elevation across watershed, in m (ELEV_STD_M_BASIN) |
| Proportion canals/pipes | Proportion of flowlines that are canals, ditches or pipes, unitless (prop_canals_pipe) |
| Percent tile drains | Estimate of percent of watershed drained by tile drains in 2012 (CPRAC_tiledrains) |
| Percent conservation tillage | Estimate of percent of watershed with conservation tillage in 2012 (CPRAC_conservation_till) |
| Long-term average precip | Mean annual precip for the watershed, using 1981-2010 record, in cm (PPT_AVG_8110) |
| Long-term average temp | Mean annual air temperature for the watershed, using 1981-2010 record, in C° (T_AVG_8110) |
| Long-term relative humidity | Mean relative humidity for the watershed, using 1961-1990 record, in percent (RH_AVG) |
| Base flow index | Base Flow Index, which is the ratio of base flow to total streamflow, in percent (BFI_AVE) |
| Percent clay | Percent of watershed with clay soils (CLAYAVE) |





| Short name | Data description (original time-series or static variable name from referenced source) |
|---|---|
| Average permeability | Average permeability, in inches per hour (PERMAVE) |
| Erosion potential (K) | K factor from Universal Soil Loss Equation, higher values = greater potential for erosion, unitless (KFACT_UP) |
| Era of first dev, watershed | Era of first development, i.e. a measure of if the area was developed a long time ago or recently. Original values from Falcone (2017) converted to decimal decade, e.g. 2000.75 means 1st development occured about 3/4 of the way through the 2000 decade (so circa 2007). Note, 1940 means 1940 or earlier. (ERA_FIRSTDEV) |
| Era of first dev, proximal zone[2] | Same as "Era of first dev, watershed" but calculated only considering the near-site riparian zone (RIP_NRSITE_ERA_FIRSTDEV) |
| Major dam density in 2013 | Number of major dams per 100 sq km across watershed in 2013 (MAJDAMS_100sqkm_2013) |
| Dam storage in 2013 | Dam storage across watershed in 2013, in acre feet per 100 sq km (NORMSTOR_sqkm_2013) |

[1]All land-use variables rounded to 1% of watershed or riparian zone area prior to calculating percent change.

[2]Variables not included in Falcone (2018) and estimated for this study using the same procedures described in Falcone (2018)



**Table 2. SSC and TSS trends from 1992 to 2012. Starting concentration, concentration change and percent change rows show: minimum - maximum (mean), rounded to two significant figures. [SSC, suspended sediment concentration; TSS total suspended solids concentration; n, count; conc, concentration; mg/L, milligrams per liter; %, percent]**

|  | SSC | TSS |
|---|---|---|
| Number of sites | 41 | 99 |
| Starting conc | 4.4 - 870 (140) | 1.1 - 270 (45) |
| Conc change (mg/L) | -410 - 72 (-51) | -83 - 40 (-9.1) |
| Conc % change | -95 - 61 (-25) | -64 - 200 (-6.3) |
| % change for upward trends | 5.8 - 61 (31) | 13 - 200 (47) |
| % change for downward trends | -95 - -11 (-43) | -64 - -8.5 (-31) |
| % change for uncertain trends | -5.1 - 10 (4.3) | -8.7 - 30 (2.7) |
| n Upward[1] trends (% of sites) | 4 (9.8%) | 20 (20%) |
| n Downward[1] trends (% of sites) | 28 (68%) | 53 (53%) |
| n Uncertain[2] trends (% of sites) | 9 (22%) | 26 (26%) |

[1]Includes likelihoods ≥ 0.70 (trend is "likely" and "somewhat likely" upward or downward)
[2]Only likelihoods < 0.70 (trend is "as likely as not" to be upward or downward, i.e. trend direction is uncertain)





**Table 3. Percent and number of sites with SSC and TSS trends for various combinations of estimates of QTC (changes in the streamflow regime) and MTC (changes in land management).**

| | SSC sites | TSS sites |
|---|---|---|
| **MTC > 0**<br>Changes in land management lead to increases in sediment | 20% (8) | 40% (40) |
| **QTC > 0**<br>Changes in the streamflow regime lead to increases in sediment | 42% (17) | 30% (30) |
| **abs(QTC) ≥ abs(MTC)**<br>Changes in the streamflow regime contribute more to the sediment trend than changes in land management | 17% (7) | 17% (17) |
| **abs(QTC) ≥ 5%**<br>Changes in the streamflow regime contribute a non-negligible amount of change to the sediment trend | 66% (27) | 57% (56) |
| **abs(QTC - sediment trend) ≤ 10%**<br>Changes in the streamflow regime account for almost the entire amount of change to the sediment trend | 10% (4) | 20% (20) |
|     **And if, abs(sediment trend) ≥ +/- 5%**<br>    In addition to abs(QTC - sediment trend) ≤ 10%, the sediment trend shows a non-negligible amount of change over the same period | 2% (1) | 10% (10) |
| **MTC & QTC have different signs**<br>The effects of changes in streamflow regime and changes in land management on the sediment trend oppose each other leading to smaller changes in sediment than either trend component alone | 51% (21) | 59% (58) |



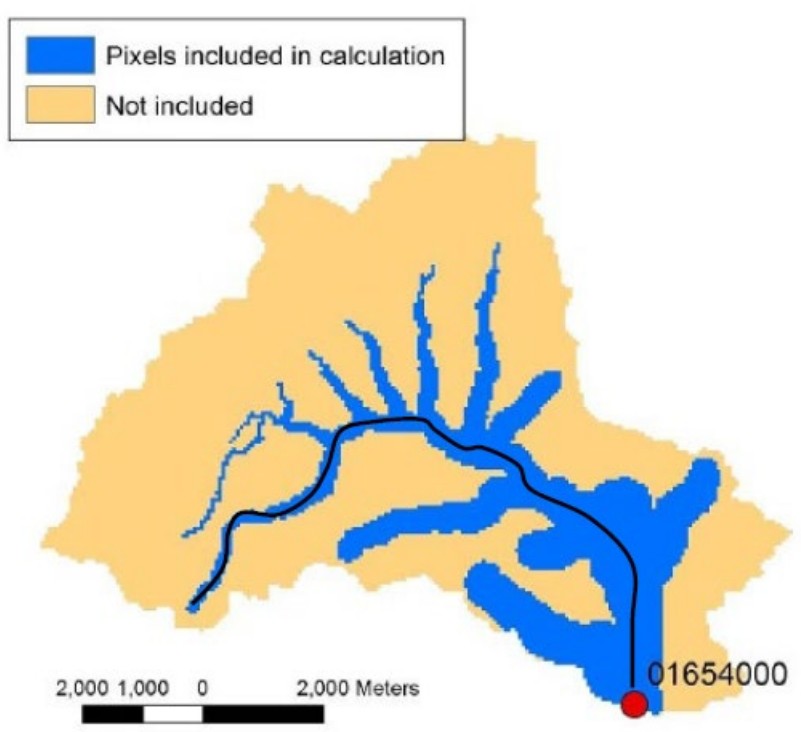

**Figure 1. Example of pixels used in near-site near-stream proximal zone calculation for site 01654000. Calculation of whole watershed values would use the tan and blue areas. Figure originally published in Falcone (2017).**





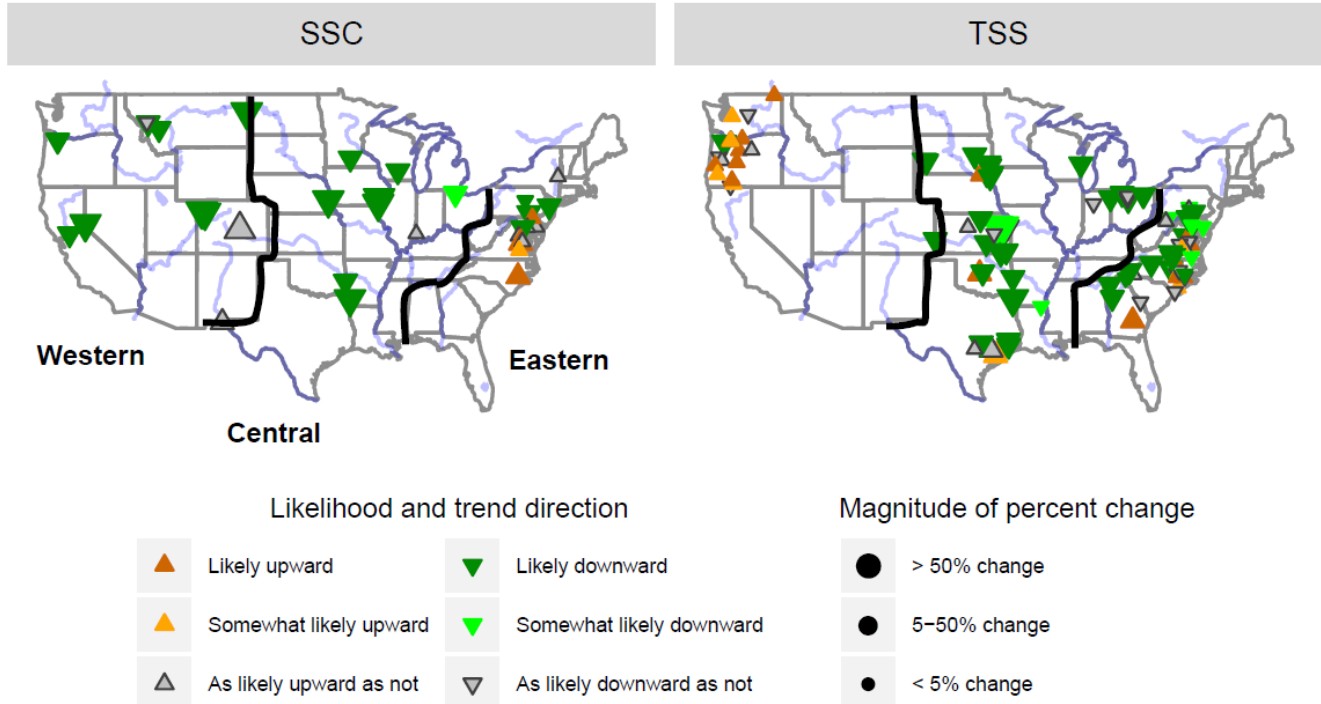

**Figure 2. SSC and TSS trends from 1992 to 2012, showing trend direction, likelihood category, and magnitude. Base map generated using ggmap in R statistical software (Kahle and Wickham, 2013).**





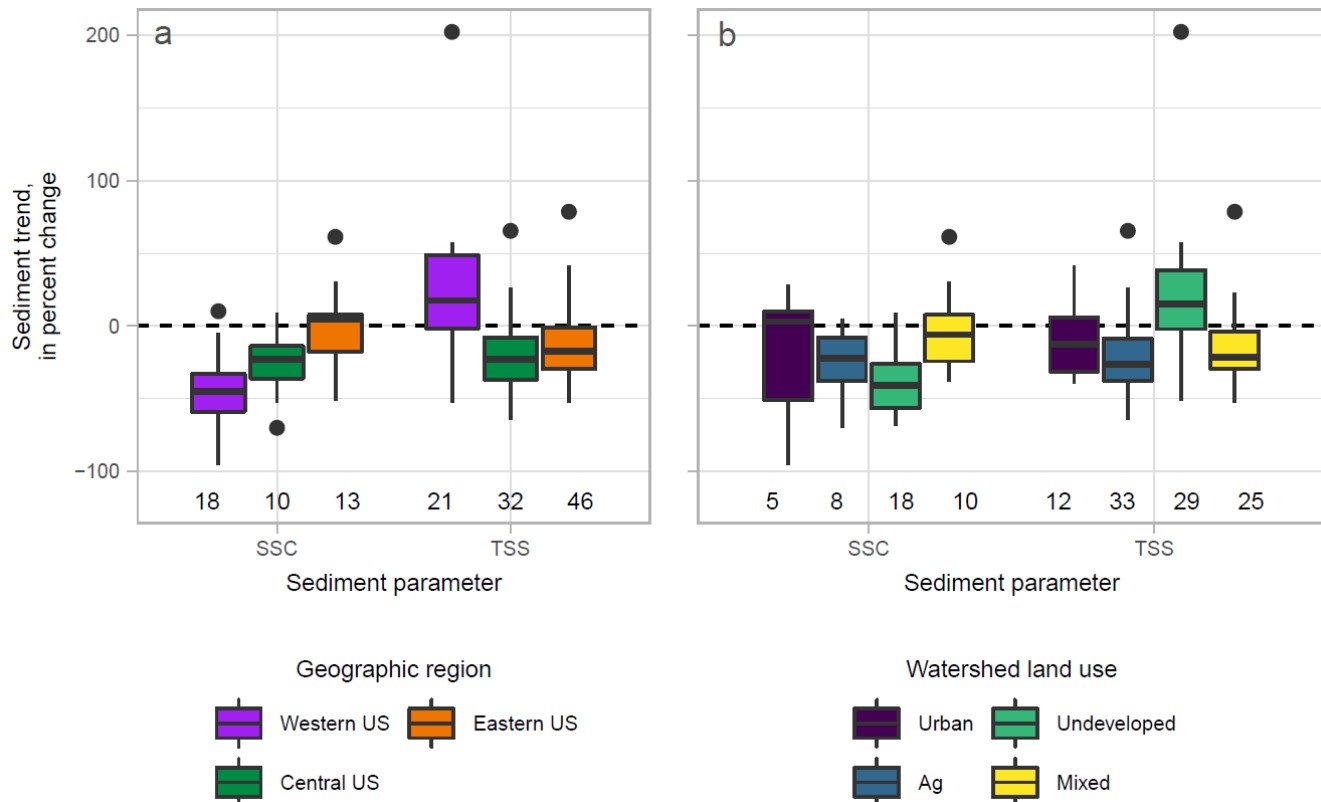

**Figure 3. SSC and TSS trends by (a) geographic regions (depicted in Figure 2) or (b) predominate watershed land use, including all sites and likelihoods. Dashed line denotes 0% change. Numbers above x-axis are site counts. For the boxplots, the top and bottom of the boxes correspond to the interquartile range (25th and 75th percentiles), top and bottom whiskers correspond to 1.5\*(interquartile range), and points are data falling beyond 1.5\*(interquartile range).**





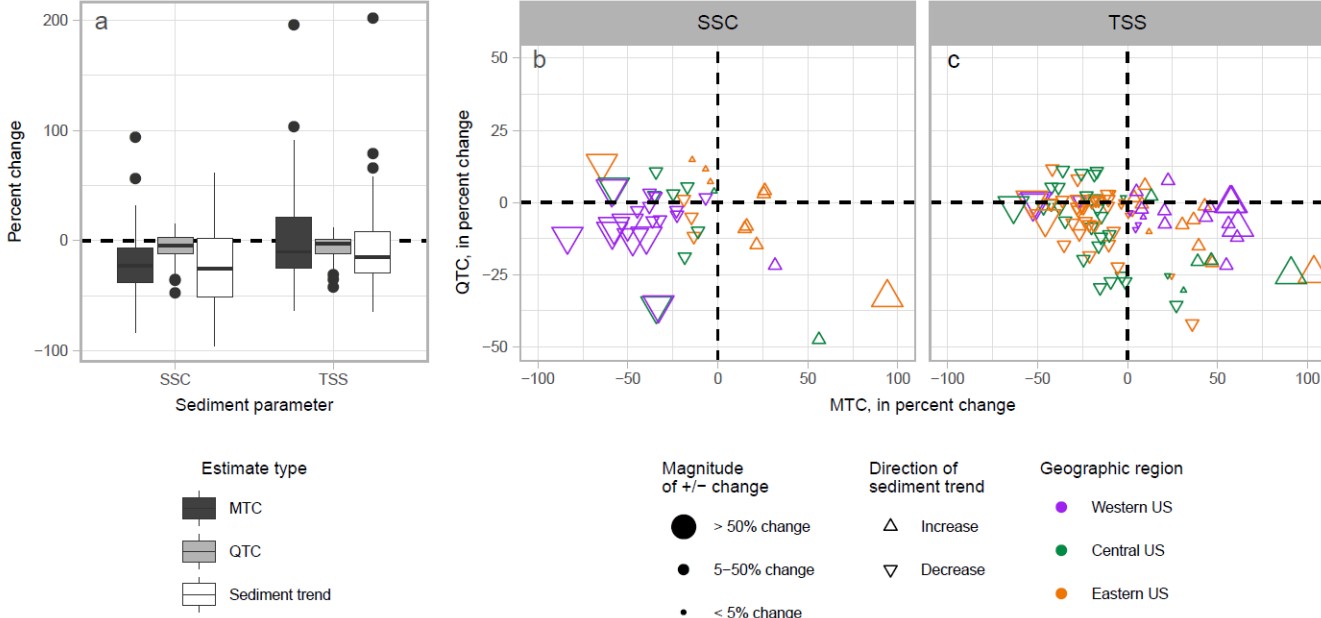

**Figure 4. (a) Boxplots of sediment trend, MTC and QTC estimates by sediment parameter. See description of boxplots in the caption of Fig. 3. (b and c) Bivariate plots of the QTC versus MTC for each site by sediment parameter, color coded by geographic regions and sized by the magnitude of sediment trend. Note, (b and c) plots exclude 1 undeveloped western US site, orBRSS0035, with 195% MTC, 7% QTC and 202% sediment trend. Recall, at a given site, sediment trend = MTC + QTC.**





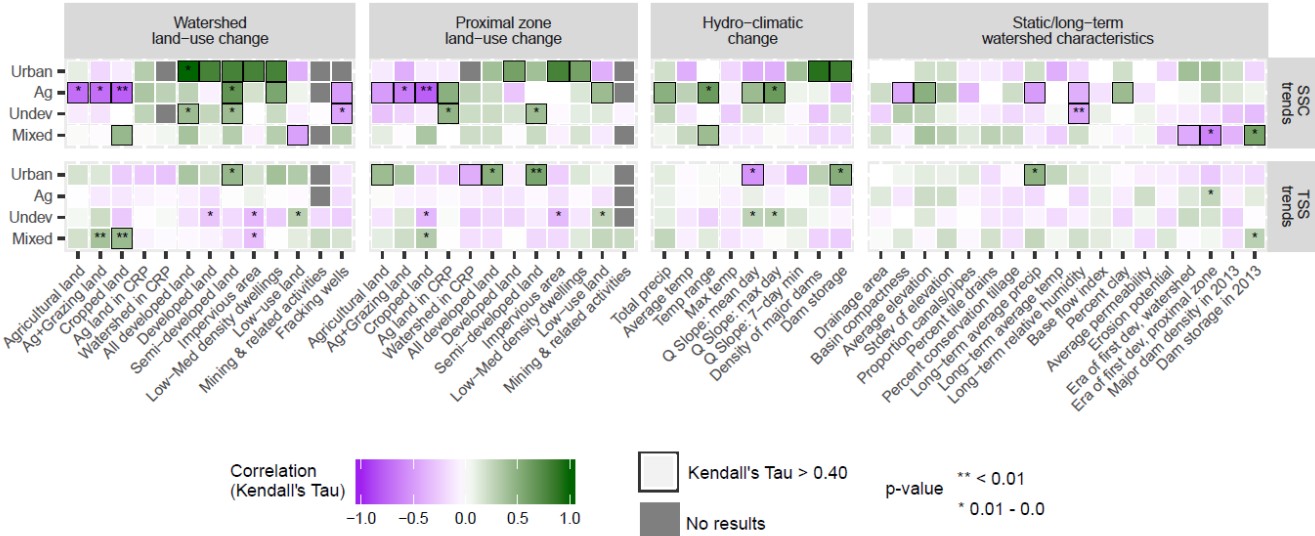

**Figure 5. Correlations between 1992-2012 sediment trends and various potential causal variables and static/long-term watershed characteristics, grouped by the 2012 land use of the contributing watershed. Note watershed and proximal zone land-use change variables and hydro-climatic change variables were correlated with sediment trends in percent change, whereas the static/long-term watershed characteristics were correlated with the sediment trends in absolute percent change.**





**Figure 6. (a and b) Sediment trend versus percent change in proportion of agricultural land in the proximal zone enrolled in CRP between 1992 and 2012. (c and d) Sediment trend versus percent of land in watershed enrolled in CRP during the trend start year (1992). Note all plots exclude site orBRSS0035 (~200% increase in TSS and 0% change in CRP).**



**Figure 7. Bivariate plots of the SSC trend (overall change in sediment concentration), MTC and QTC versus two potential causal variables: percent change in the percentage of low-medium density dwellings in the watershed and percent change per year (slope) of annual mean streamflow, for the 1992-2012 trend period. Black line is an ordinary least squares regression fit through all the data to show relationship between variables, and dashed lines indicate 0% change.**





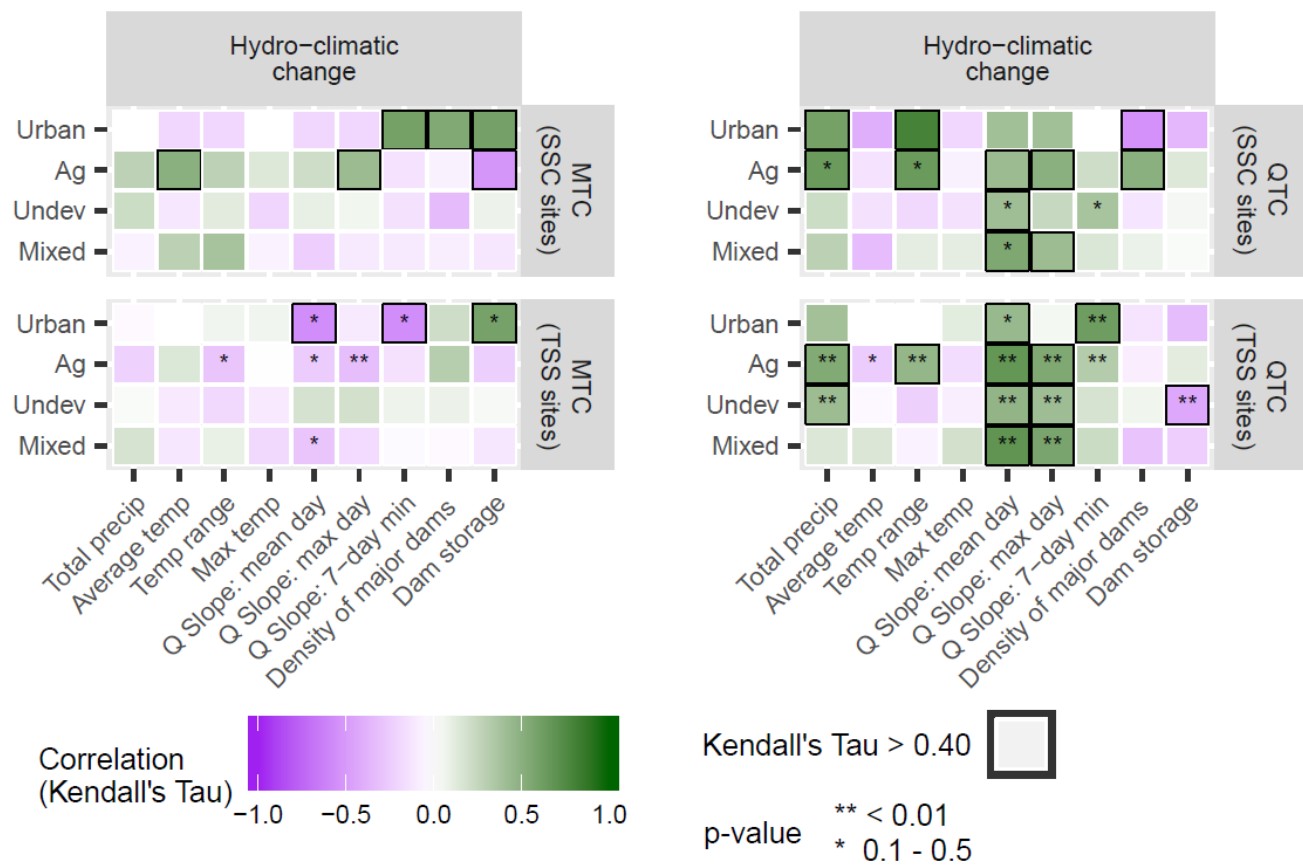

**Figure 8. Correlations between hydro-climatic change variables and MTC or QTC, by sediment parameter. Sites grouped by 2012 land use of contributing watershed.**