# Peer review of "Changing suspended sediment in United States rivers and streams: Linking sediment trends to changes in land use/cover, hydrology and climate"

_Hydrology and Earth System Sciences, 2019_

## Referee Comment (RC1) · Anonymous Referee #1 · 4 Oct 2019

Major comments Methodology

Pg. 5, ln. 13-14: Here you explain how "daily concentrations are flow normalized (FN) to remove the influence of year-to-year variability from stream...". However, the manuscript focuses on exploring potential drivers grouped into two general categories: (1) land use/management changes, and (2) streamflow regime. I am a bit confused as to how you normalize / remove the effects of flow, but at the same time attribute the changes in stream sediment to "streamflow regime". Can this be clarified? Similarly, on Pg. 6, ln 30 to 31 you give citations on how you "parse water-quality trends into

the streamflow trend component (QTC) and management trend component (MTC)." Although, I agree that citations are a convenient way to cite methodologies, I believe that most of the results and conclusions are based on this "parsing" method. Therefore, it would be important to see explicitly how methods from Choquette et al. (2019), Hirsch et al. (2018a) and Murphy and Sprague (2019) were combined, modified, and used to arrive at the QTC and MTC values. I think the current length of the manuscript is good. Therefore, this should be included in the supporting information. I believe this will make the work presented in this manuscript more transparent for future readers, rather than trying to mix and match methods from three different sources. One can then better understand how these methods were applied in this manuscript and judge whether the presented results make sense.

Pg. 5, In 23-24: "...to gauge the uncertainty of the trends, likelihood estimates of the trend direction for each site and parameter were extracted from Murphy et al. (2018)". I tried to find how this was calculated by looking up this reference. However, this appears to be a USGS dataset which points at yet another set of citations for methods. I believe the citation for the calculation of likelihood is from Oelsner et al. (2017). However, Oelsner et al. (2017) calculates a number of things. Similar to the comment above, I think the supporting information should include a section where the author lists the equations used for the most critical calculations being made (the ones the support the figures, results, and conclusions). The citations can and should still be in the manuscript. However, they are a poor substitute for trying to understand the statistical methods that were used in this manuscript. I think a short appendix where the author provides the reader with the methods used could help the reader identify the necessary methods to recreate the results presented in this manuscript. Pg. 5, In 10-29. Just curious as to why the author did not use Ryberg et al. (2013)'s SEAWAVE-Q method to separate seasonality and streamflow from the sediment concentration data? This is an available method from the USGS, so I expected it would be popular amongst other USGSers. Is WRTDS superior to SEAWAVE-Q for some reason? Furthermore, it would be important to cite Sullivan et al. (2009)'s contribution to your introduction (pg

2., ln 9-26) who compared various statistical methods for trend detection.

Cluster of increasing sediment in NW US

Pg. 1, ln. 1, title: "Declining suspended sediment in US rivers and streams" – How about the northwest coast TSS cluster that is showing a clear increase in suspended sediment. A more accurate title would be "Changing suspended sediment in United States rivers and streams…". Pg. 8, ln 13-16. Discussions about the TSS trends in northwestern US seem to need a better explanation. One reason that comes to mind is deforestation, which is arguably one of the largest contributors (in terms of land use change) to suspended sediment concentrations in rivers. A quick search on Global Forest Watch (https://www.globalforestwatch.org/) shows that there has been a decrease in tree cover in the states of Washington and Oregon since 2000. Perhaps there is a correlation between the decrease in tree cover and increase in suspended sediment concentration. Pg. 8, ln. 25 "For TSS, undeveloped sites had the largest proportion of upward trends and some of the largest increases in TSS compared to sites in other land-use categories": again this seems to beg further explanation. Not sure if the "undeveloped sites" are forested regions mainly used for timber. Pg. 8, ln 30 – 33: "Thus, the stark difference between the largely downward SSC trends and largely upward TSS trends at undeveloped sites in western US could be due to differences in the causes of changes for undeveloped sites…". OK but this is an unsatisfying explanation. The key would be to dig a bit further to help the reader understand why there are strong spatial correlations, which there appears to be from within the TSS data in Northwest US.

Pg 14, ln. 25-26 and Table 1: Should add forestry/logging to "Land-use and land-cover changes across entire watershed". This could explain the Northwest increase in TSS.

Pg. 17, ln 28-30: Your statement about many sites exhibiting a decrease in sediment should include a statement about the cluster of increase sediment trend in undeveloped NW US. Surely the remarkable pattern there deserves some recognition and further

explanation.

Outlook

Pg. 17, ln. 12: This section lays out the limitations nicely. However, what is missing is a brief outlook. How can we make better sense of these results in the future? What are the priorities for this work moving forward? Is sediment pollution going to be a problem in the future? Or do the trends suggest that this problem is solved? Give the reader your take on where this research needs to go next and what the next few decades will be like based on what your learned from your analysis and the last two decades.

Minor comments Pg. 1, ln. 23 and Pg. 17, ln. 17: You suggest that "conservation efforts" may be successful to reduce sediment runoff as lands are converted to urban and agricultural uses. These "conservation efforts" sound vague, are there any specific efforts you are speaking about. Is there any evidence, from either literature or observations, that these conservations efforts are effective? Also, Pg. 9, ln. 14: ". . . management actions on the landscape likely led to decreases in sediment concentration". Same comment here, this is a vague statement about management actions. Any ideas which management actions are effective in reducing suspended sediment concentrations? Are there many? Pg. 12, ln 5: "suggesting conservation efforts to reduce sediment runoff to streams may be successful". Can you be more specific here? Pg. 17, ln. 17: Again what efforts are you speaking of?

Pg. 3, ln. 18: Can you describe the mitigation measures that are being implement in the Conservation Reserve Program?

Pg. 3, ln 26: ". . . to characterize changes in annual mean concentrations of suspended sediment." Are the annual means a good metric to be looking at for long term trends. Annual mean concentrations can be easily skewed by a large number of low concentration values during low flow periods (e.g., in the winter when runoff is minimal across the northern US. Wouldn't one expect to have a large amount of low suspended sediment concentrations that would skew the average. Would a clearer picture of the annual sus-

pended sediment concentrations come from looking at annual median, 75% percentile or peaks concentrations (e.g., that come during the spring melt and/or high intensity short duration rainfall events)?

Pg. 8, ln 4-7: "Larger percent decreases tended to occur at sites with high concentrations in 1992 whereas the largest percent increases occurred at sites with low starting concentrations (Fig. SM-1)." There are only about 9 samples that fit this description on the SSC plot of Fig. SM-1. The rest which appears to be a cluster of samples (probably more than 9) are closer to a starting concentration of less than or equal to 60 mg/L. The point being that there are a large number of large decreases with low starting concentration as well. Therefore, this statement does not seem to accurately reflect what is presented in Fig. SM-1.

Figure 2: This is a very nice figure that should be enlarged for the Western, Central, and Eastern regions. At the continental scale it is a bit difficult to see spatial trends. Specifically, it is difficult to see the spatial distribution of triangles for TSS (especially Northwest and Eastern regions). For SCC the difficulty of seeing the spatial distribution is mainly in the Eastern regions.

Pg. 12, ln. 1: "...changes in the number of low-medium density dwellings ... had little to no effect on the streamflow regime." I find this hard to believe, would not increase in urbanization change the streamflow regime (e.g., increase rainfall-runoff response from increased paved surfaces).

Pg 12, ln. 30-32: "Previous models have suggested that changes in climate will lead to increases and decreases in sediment in particular rivers or areas of the western US." This is an ambiguous statement. I suggest to delete it or elaborate a bit more with an explanation of where increases and decreases are expected.

Pg. 13, ln. 24: "indicated that large decreases in streamflow relate to large decreases in sediment concentration".

Pg 13, ln. 27-28: "... these improvements are partially offset by human activities in the waetershed." I think this partial offset you speak of is not supported by Figure 7 because the negative correlation MTC for Q slope is very weak.

Fig. 8: Including Watershed Land-Use Change as an additional column would make this figure more insightful. I would imagine that MTC should have a stronger correlation and p-values associated with Watershed Land-Use Change.

Pg. 14, ln. 25-32: This explanation with how to interpret Fig. 5 should be when Fig. 5 is 1st introduces (i.e., Pg. 9, ln. 26, rather than 5 pages later.

Pg. 14, ln. 33: The explanation of high relative humidity leading to more vegetation and less erosion is very helpful in interpreting and understanding the results presented in Fig. 5. More explanations like these would be helpful.

Pg 15, ln. 17-20: Here you indicate that QTC and MTC usually exhibit opposite trends. Can you speculate as to what this means and why this happens?

Pg. 15, ln 25-27: Here you mention that 1 SSC and 10TSS trends had a large change in sediment and MTC near zero. Is there a reason for this? Is there a spatial pattern for these sites? What is special about these sites?

Pg. 17, ln. 19: You state here that land management was the primary contributor of changes in sediment. Can you give the average percentage? You also state that streamflow regime had a mild-to-moderate influence on sediment. Can you give the average percentage here? The purpose of this comment is to move away from a qualitative statement to a quantitative one.

Pg. 17, ln 23-25: The proximal zone results is not discussed in very much detail in the manuscript and adds little insight. I would suggest removing it from the main text and figures. Move it to the supporting information.

Fig. 5: Fracking wells is negatively correlated for SSC in undeveloped lands. Can you explain why this may be the case? I can imagine that fracking activity would require

large quantities of groundwater extraction and that this could decrease local stream baseflows, leading to higher TSS and SSC values. Instead the opposite is true, can you explain?

Fig. 7: In the 'Low-med density dwellings' column, there appears to be a number of undeveloped and mixed land-use points located along a vertical line on the zero change in low-med density dwellings. Is there an explanation for this pattern? Also, have you considered non-linear regression? Did any of the relationships exhibit non-linear dependencies?

**TREE COVER LOSS IN OREGON**   SHOW ON MAP ⚙ ℹ ⤴

From **2001** to **2018**, **Oregon** lost **1.72Mha** of tree cover, equivalent to a **18%** decrease in tree cover since **2000**, and **1.30Gt** of $CO_2$ emissions.

2000 tree cover extent | >30% tree canopy | these estimates do not take tree cover gain into account

**Fig. 1.**

**TREE COVER LOSS IN WASHINGTON** `SHOW ON MAP` ⚙ ⓘ ⌘

From **2001** to **2018**, **Washington** lost **1.54Mha** of tree cover, equivalent to a **18%** decrease in tree cover since **2000**, and **743Mt** of $CO_2$ emissions.

2000 tree cover extent | >30% tree canopy | these estimates do not take tree cover gain into account

**Fig. 2.**

---

## Referee Comment (RC2) · Anonymous Referee #2 · 14 Nov 2019

This manuscript presents an extensive data set of suspended sediment and TSS trends at 137 stream sites across the contiguous US and explores potential drivers of these changes. Overall, I think the manuscript is well written and will become a worthwhile contribution to the hydrological community. The proposed method also has the potential of being applied elsewhere. I do have some comments for the author, which I hope can help improve the manuscript.

1. On the flow-normalization trend method: It would be helpful to provide an example to guide the readers through the calculations of MTC and QTC and how the two ap-

proaches differ from each other. This essential information could be shown as Figure 1.

2. On the use of sediment concentration: Why is not sediment flux used instead? Given that both concentration and flux are assessed in the flow-normalization, why did the author choose to focus on concentration in this work?

3. Abstract: Suggest adding an opening sentence to place the work into a broader context. Also, suggesting adding 1-2 sentences to highlight the implications and relevance of the major findings.

4. P2L22: List some examples under the category deterministic approaches and empirical approaches.

5. P3L1: Be more specific on "the latter two contributions" and support this argument with literature.

6. P3L4-L23: I appreciate these thoughtful statements on the relative effects of streamflow and landscape management. However, how about efforts/practices that might affect both the streamflow regimes and landscape functioning?

7. P3L25: "suspended sediment and total suspended solid"

8. P6L8: What is the window for loess smoothing?

9. P8L30: Could you support this last sentence by showing the distribution of trends among different regions for just the undeveloped sites?

10. P10L3-L16: I appreciate these discussions by the author. However, this is not well supported by the scientific literature. Could you provide some relevant references?

11. P11L14: Any reference on these stated effects of CRP?

12. P13L14: One very relevant example on the effects of dams on sediment trend is the Conowingo Dam on Susquehanna River. There are also documented effects of

many small mill dams in the mid-Atlantic region.

13. Figure 1: I don't think this figure is necessary. You may move it to SM.

14. Table 3: I found the table with such lengthy descriptions difficult to follow. Could you convert it to a figure or shorten the descriptions?

15. Figure 2: Consider using smaller symbol to make the Eastern stations more distinct. I appreciate that the author is using the font size to represent different magnitudes, but that might be less important. Alternatively, and perhaps more conveniently, enlarge the size of the figure to be full-page so the stations can be more distinguishable.

16. Figure 3: There are outliers for many of the boxplots. What are those stations and why they have such large trends? This deserves attention from the readers and more discussion by the author.

17. Figure 5: I think this is such an important figure in the manuscript and it deserves to be made larger (say full-page) to be clearer. How about transposing this figure?

---

## Author Comment (AC1) · 11 Dec 2019

Response to Anonymous Referee #1

Thanks for the comments. I believe the suggestions will increase the clarity of the report. I've provided author responses (AR) below each Referee comment (RC).

Major comments Methodology

RC1: Pg. 5, ln. 13-14: Here you explain how "daily concentrations are flow normalized

(FN) to remove the influence of year-to-year variability from stream. . .". However, the manuscript focuses on exploring potential drivers grouped into two general categories: (1) land use/management changes, and (2) streamflow regime. I am a bit confused as to how you normalize / remove the effects of flow, but at the same time attribute the changes in stream sediment to "streamflow regime". Can this be clarified?

AR1: The flow normalization (FN) process removes the effects of year-to-year variability in streamflow but still retains the influence of systematic or progressive changes in magnitude, frequency or duration of flows over time (aka changes in the "flow regime"). All rivers have a characteristic flow regime that captures the "typical" temporal patterns of high and low flows across a year. With a stationary flow regime, some years will have higher (or lower) flows than other years due to variable weather, but this year-to-year variability in the magnitude, frequency or duration of flows remains within the expended range for the given river. For example, one year might have a higher spring flood than the next year, but this variability is expected and captured in the river's "flow regime". This is the "year-to-year" variability that is being removed during the FN process. However, nonstationary conditions lead to some (or many) parts of the flow regime to shift over time. For example, high flows may become higher or more frequent over time. This variability is a systematic change streamflow and a fundamental shift in the flow regime. The effects of these systematic changes in streamflow on water quality are still retained in the FN process, whereas the effects of year-to-year variability in streamflow are removed. Another way to think about it is that the FN process attempts to remove the "noise" introduced from variable climate (year-to-year, non-systematic, variability) but still capture effect of the "signal" from any systematic changes in streamflow on water quality. I will add text to the introduction and methods to clarify the difference between "year-to-year" variability in streamflow (noise) and systematic changes in streamflow magnitude, timing and frequency (signal) and why parsing out these types of variability help us better understanding changes water quality. Thanks for noting this point of confusion.

RC2: Similarly, on Pg. 6, ln 30 to 31 you give citations on how you "parse water-quality trends into the streamflow trend component (QTC) and management trend component (MTC)." Although, I agree that citations are a convenient way to cite methodologies, I believe that most of the results and conclusions are based on this "parsing" method. Therefore, it would be important to see explicitly how methods from Choquette et al. (2019), Hirsch et al. (2018a) and Murphy and Sprague (2019) were combined, modified, and used to arrive at the QTC and MTC values. I think the current length of the manuscript is good. Therefore, this should be included in the supporting information. I believe this will make the work presented in this manuscript more transparent for future readers, rather than trying to mix and match methods from three different sources. One can then better understand how these methods were applied in this manuscript and judge whether the presented results make sense. AR2: While was writing, I was unsure about how much "background" (previously published) method information to include in the manuscript. Based on this feedback, I see that elaborating more fully on the methods will help a reader better understand the process. I will add a section to the Supporting Information that more clearly describes how the various methods from Hirsch et al. (2018a), Choquette et al. (2019) and Murphy and Sprague (2019) all fit together.

RC3: Pg. 5, ln 23-24: "...to gauge the uncertainty of the trends, likelihood estimates of the trend direction for each site and parameter were extracted from Murphy et al. (2018)". I tried to find how this was calculated by looking up this reference. However, this appears to be a USGS dataset which points at yet another set of citations for methods. I believe the citation for the calculation of likelihood is from Oelsner et al. (2017). However, Oelsner et al. (2017) calculates a number of things. Similar to the comment above, I think the supporting information should include a section where the author lists the equations used for the most critical calculations being made (the ones the support the figures, results, and conclusions). The citations can and should still be in the manuscript. However, they are a poor substitute for trying to understand the statistical methods that were used in this manuscript. I think a short appendix where

the author provides the reader with the methods used could help the reader identify the necessary methods to recreate the results presented in this manuscript.

AR3: Good point. The appropriate reference for the uncertainty analysis is Hirsch et al. (2015), which was implemented using an updated R package (Hirsch et al. (2018b)). I can see how providing more detailed information in the Supporting Information would help a reader better understand these method details. I will add this, along with the other method information mentioned in a previous comment, to the Supporting Information. Furthermore, I will clarify in the main body of the text the primary reference for the uncertainty analysis methods.

RC4: Pg. 5, ln 10-29. Just curious as to why the author did not use Ryberg et al. (2013)'s SEAWAVE-Q method to separate seasonality and streamflow from the sediment concentration data? This is an available method from the USGS, so I expected it would be popular amongst other USGSers. Is WRTDS superior to SEAWAVE-Q for some reason? Furthermore, it would be important to cite Sullivan et al. (2009)'s contribution to your introduction (pg 2., ln 9-26) who compared various statistical methods for trend detection.

AR4: SEAWAVE-Q was designed for determining pesticide trends. The shape of the seasonal variability of pesticide concentrations in rivers is not well modeled using a sin/cosine function, which is the typical approach used in most efforts to model trends with regression equations. SEAWAVE-Q was built to better model this distinct pattern. WRTDS is, arguably, a better model for other water quality parameters such as nutrients, major ions and sediment. Additionally, WRTDS provides features that SEAWAVE-Q does not, such as flow normalization and a more flexible model fit that allows for variation in the relationships between concentration and season, time, and flow. Furthermore, WRTDS allows for the parsing of concentration trends into other "components of change", a feature not available in SEAWAVE-Q. As an aside, the trend results presented in this manuscript are from a national-scale effort (documented in Oelsner et al., 2017) and as part of that effort we also determined pesticide trends

and used SEAWAVE-Q for those analyses.

At this point, I respectfully decline the suggestion to discuss various methods for trend detection in the Introduction. Currently, the introduction focuses on methods for understanding trends and exploring potential causes. It does not cover various methods for trend detection. There are other good papers out there that explore this topic, Sullivan et al. (2009) being one of them.

Cluster of increasing sediment in NW US

RC5: Pg. 1, ln. 1, title: "Declining suspended sediment in US rivers and streams" – How about the northwest coast TSS cluster that is showing a clear increase in suspended sediment. A more accurate title would be "Changing suspended sediment in United States rivers and streams. . .".

AR5: Good point. I will update accordingly.

RC6: Pg. 8, ln 13-16. Discussions about the TSS trends in northwestern US seem to need a better explanation. One reason that comes to mind is deforestation, which is arguably one of the largest contributors (in terms of land use change) to suspended sediment concentrations in rivers. A quick search on Global Forest Watch (https://www.globalforestwatch.org/) shows that there has been a decrease in tree cover in the states of Washington and Oregon since 2000. Perhaps there is a correlation between the decrease in tree cover and increase in suspended sediment concentration.

AR6: Agreed. Adding more discussion focused on the increasing TSS cluster in the NW US would strengthen the report. Currently, this topic is given only cursory attention. However, this portion of the text (section 3.1) focuses on presenting the trend results and broad insights from geographic cluster and land use. Thus, I plan to add a discussion of the TSS cluster in the NW US to the "Land management changes" section (section 3.2) or the "Importance of location section (section 3.4) of the manuscript.

I did not include deforestation in the correlation analysis because temporally consistent estimates of forest cover and timber harvesting are not available back to 1992. To my knowledge the earliest spatially and temporally consistent data on forest cover begins in 2001 (https://www.mrlc.gov/national-land-cover-database-nlcd-2016) and timber begins in 1999 (https://pubs.er.usgs.gov/publication/ds948). However, I agree that even without a temporal perspective equivalent to the other variables in table 1, this is still plenty of opportunity to bring in some quantitative information about forest and timber changes and relate them to changes in sediment. To that end, I plan to consider current (~2012, coinciding with trend end year) and more recent changes (1999 onward) in forest cover and timber conditions to see if they help explain some of the sediment changes observed in this study.

RC7: Pg. 8, ln. 25 "For TSS, undeveloped sites had the largest proportion of upward trends and some of the largest increases in TSS compared to sites in other land-use categories": again this seems to beg further explanation. Not sure if the "undeveloped sites" are forested regions mainly used for timber.

AR7: This is a possible explanation for the increasing TSS trends at undeveloped sites. These "undeveloped sites" may certainly include timber harvesting. "Undeveloped" is defined as a lack of agricultural and urban land uses. I will explore this line of inquiry further as described in above bullet. Any expanded discussion on this topic will be added to section 3.2 ("Land management changes") or section 3.4 ("Importance of location section") of the manuscript.

RC8: Pg. 8, ln 30 – 33: "Thus, the stark difference between the largely downward SSC trends and largely upward TSS trends at undeveloped sites in western US could be due to differences in the causes of changes for undeveloped sites...". OK but this is an unsatisfying explanation. The key would be to dig a bit further to help the reader understand why there are strong spatial correlations, which there appears to be from within the TSS data in Northwest US.

AR8: I included this sentence in the manuscript as a way to set up of the following sentence, "Other contributing factors could include differences in the suspended particle-size distributions being characterized by SSC and TSS and different regions having different underlying geology." I was trying to make the point that there may be other reasons that SSC and TSS trends differ apart from differences in landscape changes at these sites (such as TSS sites having increases in deforestation and SSC sites not). For example, SSC and TSS use different analytical procedures to determine suspended sediment. Also, there may be important differences in basin characteristics for SSC sites versus TSS sites. These two considerations mean that an actual change on the landscape (such as increased deforestation) may not affect SSC and TSS the same way. I will update this part of the text to clarify.

RC9: Pg 14, ln. 25-26 and Table 1: Should add forestry/logging to "Land-use and land-cover changes across entire watershed". This could explain the Northwest increase in TSS.

AR9: See above bullet. While there is not spatially and temporally consistent forestry/logging data back to 1992 there are some sources of information I can bring in to illuminate this important potential driver of change.

RC10: Pg. 17, ln 28-30: Your statement about many sites exhibiting a decrease in sediment should include a statement about the cluster of increase sediment trend in undeveloped NW US. Surely the remarkable pattern there deserves some recognition and further explanation.

AR10: Good point. I will update the conclusion accordingly.

Outlook RC11: Pg. 17, ln. 12: This section lays out the limitations nicely. However, what is missing is a brief outlook. How can we make better sense of these results in the future? What are the priorities for this work moving forward? Is sediment pollution going to be a problem in the future? Or do the trends suggest that this problem is solved? Give the reader your take on where this research needs to go next and what

the next few decades will be like based on what your learned from your analysis and the last two decades.

AR11: Agreed, adding this type of discussion would improve the manuscript by connecting the results presented here to larger issues. I will update the manuscript to include a paragraph that touches on these topics.

Minor comments

RC12: Pg. 1, ln. 23 and Pg. 17, ln. 17: You suggest that "conservation efforts" may be successful to reduce sediment runoff as lands are converted to urban and agricultural uses. These "conservation efforts" sound vague, are there any specific efforts you are speaking about. Is there any evidence, from either literature or observations, that these conservations efforts are effective? Also, Pg. 9, ln. 14: "...management actions on the landscape likely led to decreases in sediment concentration". Same comment here, this is a vague statement about management actions. Any ideas which management actions are effective in reducing suspended sediment concentrations? Are there many? Pg. 12, ln 5: "suggesting conservation efforts to reduce sediment runoff to streams may be successful". Can you be more specific here? Pg. 17, ln. 17: Again what efforts are you speaking of?

AR12: I use the terms "conservation efforts" and "management actions" in a general sense throughout the manuscript. What I mean by these terms are any actions taken in the watershed that would shift the concentration-discharge (C-Q) relationship over time. Shifts in C-Q relationships are one tool watershed managers use to evaluate progress of conservation and management efforts on water quality. For example, the Delaware River Basin Commission uses changes in the C-Q relationship as a way to detect "measurable change to existing water quality" (https://www.nj.gov/drbc/library/documents/LowerDel_EWQrpt_2016/LDel_EWQrpt_2016_entire.pdf). Also, Moatar et al. (2017) used C-Q relationships to gauge the effect of changes in point sources across streams in Europe. However, I see the point in providing some

more concrete examples. To this end, I will add some references to the paper to further elaborate on the potential conservation efforts and management actions that could lead to changes in the C-Q relationship.

Moatar et al., 2017, Elemental properties, hydrology, and biology interact to shape concentration-discharge curves for carbon, nutrients, and sediment, and major ions. Water Resources Research 53, 1270–1287, doi:10.1002/2016WR019635.

RC13: Pg. 3, ln. 18: Can you describe the mitigation measures that are being implement in the Conservation Reserve Program?

AR13: The Conservation Reserve Program (CRP) pays farmers to not farm environmentally sensitive land and instead plant environmentally beneficial plants. This can include buffer strips and wetlands. Will add text to explain.

RC14: Pg. 3, ln 26: "…to characterize changes in annual mean concentrations of suspended sediment." Are the annual means a good metric to be looking at for long term trends. Annual mean concentrations can be easily skewed by a large number of low concentration values during low flow periods (e.g., in the winter when runoff is minimal across the northern US. Wouldn't one expect to have a large amount of low suspended sediment concentrations that would skew the average. Would a clearer picture of the annual suspended sediment concentrations come from looking at annual median, 75% percentile or peaks concentrations (e.g., that come during the spring melt and/or high intensity short duration rainfall events)?

AR14: The analysis presented in this manuscript does not use a mean calculated directly from the observed samples (which would lead to "under-weighting" concentrations during high flow periods). Instead it uses modeled mean annual concentrations that are derived using a weighted regression equation that includes terms for time, season and discharge. Furthermore, the data used to the calibrate the model were screened for high flow samples so that we can be more confident that the mean annual concentrations are reflecting concentrations at high flows as well as low and moderate flows. Thus, the mean annual concentrations used in this analysis are accounting for concentrations at high flows and are not underweighting these conditions during the estimation of mean concentrations. I think adding more specific information about the methods and the relevant equations to the Supporting Information, as previously suggested by referee 1, will help to better explain this.

Additionally, the MTC and QTC approach presented in the manuscript provides a novel way to explore the influence of changing flow conditions on suspended sediment. We know that for constituents like sediment, most of the transport occurs during high flow events. Thus, changes in the flow regime, particularly at high flows, is very important to how sediment changes over time. This kind of information would be difficult, or impossible, in glean from exploring the effect of changes in mean streamflow on sediment. Thus, I argue that the MTC and QTC approach in WRTDS provides an ideal way to explore changes in mean annual concentrations because of the approach's ability to identify the effects of different types of flow changes, including changes in flow magnitudes, frequencies, and timing, on sediment. Again, I think elaborating on the methods in the Supporting Information will make this clearer to an interested reader.

RC15: Pg. 8, ln 4-7: "Larger percent decreases tended to occur at sites with high concentrations in 1992 whereas the largest percent increases occurred at sites with low starting concentrations (Fig. SM-1)." There are only about 9 samples that fit this description on the SSC plot of Fig. SM-1. The rest which appears to be a cluster of samples (probably more than 9) are closer to a starting concentration of less than or equal to 60 mg/L. The point being that there are a large number of large decreases with low starting concentration as well. Therefore, this statement does not seem to accurately reflect what is presented in Fig. SM-1.

AR15: I agree. I think the better point to be made here is that decreases in sediment occurred at sites with low to very high concentrations in 1992, whereas increases in sediment did not occur at sites with high starting concentrations. I will update the text to clarify.

RC16: Figure 2: This is a very nice figure that should be enlarged for the Western, Central, and Eastern regions. At the continental scale it is a bit difficult to see spatial trends. Specifically, it is difficult to see the spatial distribution of triangles for TSS (especially Northwest and Eastern regions). For SCC the difficulty of seeing the spatial distribution is mainly in the Eastern regions.

AR16: I will rectify this issue by adding transparency to the symbols, eliminating the different symbol sizes, and/or enlarging all or a portion of the maps.

RC17: Pg. 12, ln. 1: "...changes in the number of low-medium density dwellings ... had little to no effect on the streamflow regime." I find this hard to believe, would not increase in urbanization change the streamflow regime (e.g., increase rainfall-runoff response from increased paved surfaces).

AR17: In hindsight, I agree this statement about "little to no effect on the streamflow regime" is too strong and poorly worded. I meant to make the point that the relationship between low-medium density dwellings and the QTC is much more muted compared to the MTC indicating changes in the low-medium density dwelling appear to affect overall sediment concentrations more strongly via the C-Q relationship compared to the flow regime. The QTC is describing the amount of change in sediment attributed to changes in the flow regime; there may have been considerable changes in the flow regime at many of these sites due to changes in low-medium density dwellings, but these changes did not affect sediment concentrations with the same magnitude that changes in the C-Q relationships did. I will update the text to clarify.

RC18: Pg 12, ln. 30-32: "Previous models have suggested that changes in climate will lead to increases and decreases in sediment in particular rivers or areas of the western US." This is an ambiguous statement. I suggest to delete it or elaborate a bit more with an explanation of where increases and decreases are expected.

AR18: Agreed. I will delete.

RC19: Pg. 13, ln. 24: "indicated that large decreases in streamflow relate to large decreases in sediment concentration".

AR19: I'm not entirely sure what the issue is with this quoted portion of the sentence. I will update the whole sentence to: "However, the well-defined positive relationship between mean daily streamflow and the QTC indicates that large decreases in streamflow relate to large decreases in sediment concentration (Fig. 7).

RC20: Pg 13, ln. 27-28: "...these improvements are partially offset by human activities in the waetershed." I think this partial offset you speak of is not supported by Figure 7 because the negative correlation MTC for Q slope is very weak.

AR20: Upon further review, I agree, I don't think these figures are the best to make that point (figures 4b and 4c would be better). I'm going to re-scope this paragraph so it's more about the ways that the QTC and MTC relate to different types of changes on the landscape and in streamflow. Basically, the sediment trend is largely driven by the MTC and much less influenced by the QTC. Furthermore, the MTC is more strongly related to changes on the landscape (middle-left panel of figure 7) whereas QTC is more strongly related to changes in streamflow (bottom-right panel of figure 7). Thanks for point this out.

RC21: Fig. 8: Including Watershed Land-Use Change as an additional column would make this figure more insightful. I would imagine that MTC should have a stronger correlation and p-values associated with Watershed Land-Use Change.

AR21: Good suggestion. I will update the figure to include the watershed land-use change variables and add some text to discuss it. With this addition, I may or may not keep figure 7. It seems the point about what effects the MTC and what effects the QTC can be made with the expanded figure 8.

RC22: Pg. 14, ln. 25-32: This explanation with how to interpret Fig. 5 should be when Fig. 5 is 1st introduces (i.e., Pg. 9, ln. 26, rather than 5 pages later.

AR22: When figure 5 is first introduced (on page 9), this is in the section 3.2 "Land management changes" where I discuss the watershed land-use change variables and correlations. Page 14 is in the section 3.4 "Importance of location" and in this section I'm discussing the portion of fig 5 that has the static and long-term watershed characteristics. I think moving it up to section 3.2 would muddle the focus of that section. However, I will consider creating a separate figure for the "Static/long-term watershed characteristics" portion of the heat map – that might help clarify the different focuses of these correlations.

RC23: Pg. 14, ln. 33: The explanation of high relative humidity leading to more vegetation and less erosion is very helpful in interpreting and understanding the results presented in Fig. 5. More explanations like these would be helpful.

AR23: Agreed. I will work in more concrete examples like this throughout the manuscript.

RC24: Pg 15, ln. 17-20: Here you indicate that QTC and MTC usually exhibit opposite trends. Can you speculate as to what this means and why this happens?

AR24: An example of this effect at a single site is given in Murphy and Sprague (2019). In that paper we showed that SSC concentrations decreased by 70% at a site on the Skunk River in Iowa, USA between 1982 and 2012. This change was attributed to a -90% MTC and 20% QTC. In Iowa the rate of soil erosion decreased in the 1980s and 1990s and this was attributed to taking erodible land out of production. These practices appear to have led to a shift in the C-Q relationship which was ultimately expressed in a -90% MTC. However, during this time there were also increases in streamflow during the spring and summer which likely lead to increased mobilization of sediment over this period and the positive 20% QTC shown at this site. Taken together, conservation practices at this site may have led to decreased sediment transport which would have been even greater if there hadn't been concurrent increases in streamflow. I can see how giving some more examples and elaborating on these "opposing effects" would

greatly help the reader. Thanks for the suggestion.

Murphy and Sprague, 2019, Water-quality trends in US rivers: Exploring effects from streamflow trends and changes in watershed management. Science of the Total Environment 656, 645-658. https://doi.org/10.1016/j.scitotenv.2018.11.255

RC25: Pg. 15, ln 25-27: Here you mention that 1 SSC and 10TSS trends had a large change in sediment and MTC near zero. Is there a reason for this? Is there a spatial pattern for these sites? What is special about these sites?

AR25: I pointed out these 11 sites because I wanted to emphasize how uncommon it was for changes in sediment to be totally driven by changes in the flow regime alone (no concurrent changes in the C-Q relationship). However, I see now that I could elaborate a bit about these sites. I will explore these sites in more detail to see if there are any spatial patterns or particular land use changes associated with them.

RC26: Pg. 17, ln. 19: You state here that land management was the primary contributor of changes in sediment. Can you give the average percentage? You also state that streamflow regime had a mild-to-moderate influence on sediment. Can you give the average percentage here? The purpose of this comment is to move away from a qualitative statement to a quantitative one.

AR26: Good point. I will update this portion of the text with the appropriate numbers.

RC27: Pg. 17, ln 23-25: The proximal zone results is not discussed in very much detail in the manuscript and adds little insight. I would suggest removing it from the main text and figures. Move it to the supporting information.

AR27: Agreed. That analysis was underwhelming and does not add much to the overall interpretation. I will move to Supporting Information which will free up more space in the manuscript for some additional discussion.

RC28: Fig. 5: Fracking wells is negatively correlated for SSC in undeveloped lands. Can you explain why this may be the case? I can imagine that fracking activity would

require large quantities of groundwater extraction and that this could decrease local stream baseflows, leading to higher TSS and SSC values. Instead the opposite is true, can you explain?

AR28: Thank you for this comment. I went back and reviewed the bivariate plots for each of the correlations reported in Fig. 5. For SSC in undeveloped lands, there are 4 sites with large changes in fracking and one of which had a much higher % change in fracking than the others. This single site is leveraging the whole relationship. Also, it appears a similar thing is happening with changes in "Mining and related activities" (2 sites with large changes are dictating the relationship). Thus, I'm going to drop these two variables from the analysis since they are skewing the correlations and do not provide much insight. Again, thank you for noting this.

RC29: Fig. 7: In the 'Low-med density dwellings' column, there appears to be a number of undeveloped and mixed land-use points located along a vertical line on the zero change in low-med density dwellings. Is there an explanation for this pattern? Also, have you considered non-linear regression? Did any of the relationships exhibit nonlinear dependencies?

AR29: The cluster of undeveloped and mixed land-use sites along the vertical axis are sites that did not see any change in the percent of watershed with low-medium density dwellings but did have a change in sediment concentration. For these sites, something other than a change in low-medium density dwellings (since there was no change) effected sediment concentration. I will update the text to make this clearer.

I did not consider non-linear regression; however, I am using Kendall's tau for the correlation analysis which is a rank-based, non-parametric method for assessing bivariate relationships. I chose Kendall's tau over Pearson's r because most of the data do not follow a normal distribution. Using Kendall's tau also does not require the assumption of linearity, just that the relationship be monotonic.

435, 2019.

---

## Author Comment (AC2) · 11 Dec 2019

Response to Referee #2

RC: This manuscript presents an extensive data set of suspended sediment and TSS trends at 137 stream sites across the contiguous US and explores potential drivers of these changes. Overall, I think the manuscript is well written and will become a worthwhile contribution to the hydrological community. The proposed method also has the potential of being applied elsewhere. I do have some comments for the author,

which I hope can help improve the manuscript.

AR: Thank you for the supportive and constructive comments on my manuscript. I've provided author respondes (AR) to the referee comments (RC) below.

RC1. On the flow-normalization trend method: It would be helpful to provide an example to guide the readers through the calculations of MTC and QTC and how the two approaches differ from each other. This essential information could be shown as Figure 1.

AR1: Referee 1 also requested additional information and explanation of the MTC and QTC methodology and suggested adding this information to the Supporting Information. Showing how the methods are applied at a specific site is another interesting suggestion. I will spend some time thinking about the best way to incorporate additional, clarifying information about the methods – either in the Supporting Information or in the manuscript with a figure.

RC2. On the use of sediment concentration: Why is not sediment flux used instead? Given that both concentration and flux are assessed in the flow-normalization, why did the author choose to focus on concentration in this work?

AR2: I went back and forth about this choice prior to beginning the analysis. Ultimately, I decided to go with sediment concentration because my primary goal of this analysis was to explore potential drivers of change. Since sediment loads are very closely related to streamflow, I thought I would be better able to identify the influence of other changes, such as land use and climate, if I used concentration (better able to get the "signal" out of the "noise" using concentration as opposed to load). I decided to only go with concentration, as opposed to concentration and loads, to keep the manuscript digestible. I suspect many of the conclusions will be similar between concentration and load because streamflow is typically positively related to both concentration and load (so increases in Q are likely to lead to increases in concentrations and loads).
RC3. Abstract: Suggest adding an opening sentence to place the work into a broader context. Also, suggesting adding 1-2 sentences to highlight the implications and relevance of the major findings.

AR3: Ok will do.

RC4. P2L22: List some examples under the category deterministic approaches and empirical approaches.

AR4: Good point, I will update.

RC5. P3L1: Be more specific on "the latter two contributions" and support this argument with literature.

AR5: I will enhance this paragraph to better support these ideas.

RC6. P3L4-L23: I appreciate these thoughtful statements on the relative effects of streamflow and landscape management. However, how about efforts/practices that might affect both the streamflow regimes and landscape functioning?

AR6: I agree that there are plenty of efforts/practices that affect both streamflow regimes and landscape functioning. I plan to dig into that more throughout the entire manuscript but will add some discussion on that here in the introduction as well.

RC7. P3L25: "suspended sediment and total suspended solid"

AR7: Will add.

RC8. P6L8: What is the window for loess smoothing?

AR8: Loess smoothing was applied in R using the loess() function with the span argument set to 0.75. Meaning 75% of the points are used in each window and these points have tricubic weight. Will update text to clarify.

RC9. P8L30: Could you support this last sentence by showing the distribution of trends among different regions for just the undeveloped sites?

AR9. Referee 1 also noted this sentence as being vague. I plan to drop this sentence ("Thus, the stark difference between the largely downward SSC…."). Most of the undeveloped sites are in the Western US. Site counts for the other geographic regions are too small to gain much insight. For SSC sites, there are 12, 1, and 2 sites in the Western, Central, and US regions. For TSS sites, there are 18, 5, and 6 sites, respectively.

RC10. P10L3-L16: I appreciate these discussions by the author. However, this is not well supported by the scientific literature. Could you provide some relevant references?

AR10: I respectfully disagree. It is well supported that TSS determinations are more uncertain that SSC determinations, and typically biased low. TSS determinations tend to result in a "sediment deficient" subsample based on the techniques used to retrieve a subsample from the original sample for analysis. These issues become more severe with increases in the proportion of sand-sized sediment in a sample. See method comparison by Gray et al (2000). While I discuss Gray et al (2000) in other places in the manuscript, I see that I did not include it in this section. I will rectify that issue.

Additionally, many studies have shown the preferential settling of coarser material as streamflow slows. With respect to conservation practices, White et al. (2007) showed that forested filter strips are efficient at removing coarse-textured sediment (> 20 um in diameter) but that small particles (<2 um, generally clay and smaller) are not affected. Lee et al. (2000) found that trapping efficiencies varied depending on the vegetation type used in vegetative buffers but were highest for coarse sediment. Meyer et al (1995) found that grass hedges trapped nearly all sand-sized sediment but allowed silt and clay-sized sediment through. Bimbino et al. (2008) found decreases in sediment size over a reach that had 3 check dams. I do agree this section of the manuscript lacks supporting references, so thank you for that comment. I will update the manuscript with appropriate references, such as the ones described above.

Gray, J.R., Glysson, G.D., Turcios, L.M., and Schwarz, G.E.: Comparability of

Suspended-Sediment Concentration and Total Suspended Solids Data. U.S. Geological Survey Water-Resources Investigations Report 00-4191, 2000.

White et al., 2007, Sediment retention by forested filter strips in the Piedmont of Georgia. Journal of Soil and Water Conservation 62, no. 6., 453-463.

Lee et al., 2000, Multispecies Riparian Buffers Trap Sediment and Nutrients during Rainfall Simulations. Journal of Environmental Quality 29, n. 4., 1200-1205.

Meyer et al., 1995, Sediment-trapping effectiveness of stiff-grass hedges. Transactions of the American Society of Agricultural and Biological Engineers 38(3): 809-815.

Bombino et al., 2008, Sediment size variation in torrents with check dams: Effects on riparian vegetation. Ecological Engineering 32, 166-177.

RC11. P11L14: Any reference on these stated effects of CRP?

AR11: Often process-based watershed models (such as SWAT) are used to assess the effectiveness of conservation practices on water quality, for example see the US Department of Agriculture's Conservation Effects Assessment Project (https://www.nrcs.usda.gov/Internet/FSE_DOCUMENTS/nrcseprd889806.pdf). However, identifying these effects empirically has proven challenging. To my knowledge, no one has assessed the influence of CRP on sediment transport nationally (some studies have been done for nutrients, see Sprague and Gronberg (2012)). Studies completed at individual basins give a mixed story. Davie and Lant (1994) found CRP enrollment influenced sediment erosion rates but not sediment loads downstream. They also suggest that the location of CRP near the stream might be important for effecting downstream sediment load. Support for this idea is shown in figure 6b. Lizotte et al. (2012) found decreases in sediment in an oxbow lake related to the implementation of best management practices and CRP enrollment in the surrounding drainages. Cullum et al. (2010) found the conversion of cropped land into forested CRP land in the drainage surrounding an oxbow lake reduced the sediment load entering the lake by an order

of magnitude. I will enhance this section of the report by elaborating on the documented effects of CRP in individual watersheds and discuss the difficulty of gauging these effects on a national scale.

Davie, D.K., and Lant, C.L., 1994, The effect of CRP enrollment on sediment loads in two southern Illinois streams. Journal of Soil and Water Conservation 49(4), 407-412.

Lizotte, et al., 2012, Water quality monitoring of an agricultural watershed lake: the effectiveness of agricultural best management practices. Transactions on Ecology and the Environment 160, doi:10.2495/DN120251.

Cullum, et al., 2010, Effects of Conservation Reserve Program on Runoff and Lake Water Quality in an Oxbow Lake Watershed. Journal of International Environmental Application and Science 5, (3): 318-328.

RC12. P13L14: One very relevant example on the effects of dams on sediment trend is the Conowingo Dam on Susquehanna River. There are also documented effects of many small mill dams in the mid-Atlantic region.

AR12: Agreed. It was surprising the effects of dams were not more pronounced in this study. The manuscript provides several reasons why this may be the case.

RC13. Figure 1: I don't think this figure is necessary. You may move it to SM.

AR13: Agreed. I will be moving the analysis pertaining to the riparian land-use change to the Supporting Information.

RC14. Table 3: I found the table with such lengthy descriptions difficult to follow. Could you convert it to a figure or shorten the descriptions?

AR14: I respectfully decline this suggestion. I am unsure how this table could be converted to a figure and the descriptions are about as concise as I can make them. The bolded portion of the table provide the information in a succinct format; the descriptions are provided so that a reader can gain a better understanding of how to interpret the

magnitude and direction of MTC and QTC estimates.

RC15. Figure 2: Consider using smaller symbol to make the Eastern stations more distinct. I appreciate that the author is using the font size to represent different magnitudes, but that might be less important. Alternatively, and perhaps more conveniently, enlarge the size of the figure to be full-page so the stations can be more distinguishable.

AR15: Referee 1 also had issue with Figure 2 and the clustering of sites. I will rectify this issue by adding transparency to the symbols, eliminating the different symbol sizes, or enlarging all or a portion of the maps.

RC16. Figure 3: There are outliers for many of the boxplots. What are those stations and why they have such large trends? This deserves attention from the readers and more discussion by the author.

AR16: My goal for this paper is to present a national perspective on changes in sediment concentration since 1992 across the US. Thus, I chose not to explore and elaborate on these sites with outlier changes in sediment since these likely present unique situations.

RC17. Figure 5: I think this is such an important figure in the manuscript and it deserves to be made larger (say full-page) to be clearer. How about transposing this figure?

AR17: My plan is to move the riparian land-use change analysis and results to the Supporting Information. Doing this will remove the riparian land-use change correlations from Fig 5 and will allow more space for what remains. I will also explore transposing the figure.